# Eye-specific differences in active zone addition during synaptic competition in the developing visual system

Chenghang Zhang, Tarlan Vatan, Colenso M Speer*

Department of Biology, University of Maryland, College Park, United States

## eLife Assessment

This is a **valuable** analysis of STORM data that characterizes the clustering of active zones in retinogeniculate terminals across ages and in the absence of retinal waves. The design makes it possible to relate fixed time point structural data to a known outcome of activity-dependent remodeling. The latest revision has tempered the causal claims made in previous versions. The result provides **solid** structural support for the hypotheses regarding how activity influences the clustering of these synapses.

**\*For correspondence:**
cspeer@umd.edu

**Competing interest:** The authors declare that no competing interests exist.

**Abstract** Spatially clustered synaptic inputs enable local dendritic computations important for learning, memory, and sensory processing. In the mammalian visual system, individual retinal ganglion cell axons form clustered terminal boutons containing multiple active zones onto relay cell dendrites in the dorsal lateral geniculate nucleus. This mature architecture arises through the addition of release sites, which strengthens selected afferents while weaker inputs are pruned. Following eye-opening, spontaneous activity and visual experience promote synaptic refinement and bouton clustering after binocular inputs have segregated. However, anatomical changes in release site addition and spatial patterning during earlier stages of eye-specific competition are not well understood. To investigate this, we examined the spatial organization of eye-specific active zones in wild-type mice and a mutant line with disrupted cholinergic retinal waves. Using volumetric super-resolution single-molecule localization microscopy and electron microscopy, we found that individual retinogeniculate boutons begin forming multiple nearby presynaptic active zones during the first postnatal week. Both eyes generate these 'multi-active-zone' (mAZ) inputs throughout refinement, but the dominant eye forms more numerous mAZ contacts, each with more active zones and larger vesicle pools. At the height of competition (postnatal day 4), the non-dominant-eye projection adds many single-active-zone synapses. Mutants with abnormal cholinergic retinal waves still form mAZ inputs but develop fewer synapses overall and show reduced synaptic clustering in projections from both eyes. Together, these findings reveal eye-specific differences in release site addition that correlate with axonal segregation outcomes during retinogeniculate refinement.

## Introduction

A key mechanism for neuronal signal processing is the formation of spatially clustered synapses that facilitate local computations within individual dendrites (*Rall, 1962*; *Mel, 1991*; *Poirazi and Mel, 2001*). Through biophysical signal integration mechanisms, neighboring synaptic inputs play critical computational roles in learning, memory, and sensory processing underlying cognition and behavior (*Mel et al., 2017*; *Kastellakis and Poirazi, 2019*; *Winnubst and Lohmann, 2012*; *Leighton and Lohmann, 2016*). During circuit development, both spontaneous and sensory-driven neural activity

regulate synaptic clustering, stabilizing some synapses and eliminating others to establish mature connectivity patterns (*Winnubst and Lohmann, 2012*; *Leighton and Lohmann, 2016*; *Kirchner and Gjorgjieva, 2022*).

The refinement of retinal inputs to the dorsal lateral geniculate nucleus (dLGN) of the thalamus is a model example of activity-dependent synaptic clustering (*Liang and Chen, 2020*). Over development, boutons within the terminal arbors of individual retinogeniculate axons cluster progressively (*Bickford et al., 2010*; *Monavarfeshani et al., 2018*; *Hong et al., 2014*; *Mason, 1982*), creating multiple synaptic contacts onto individual postsynaptic targets (*Hamos et al., 1987*; *Hammer et al., 2015*; *Morgan et al., 2016*). Concurrently, developing retinal ganglion cell (RGC) boutons add presynaptic release sites such that some mature retinogeniculate terminals contain several dozen active zones (AZs) (*Morgan et al., 2016*; *Budisantoso et al., 2012*). The formation of clustered boutons with multiple release sites subserves a critical 'driver' function of retinogeniculate input (*Sherman and Guillery, 1998*; *Guillery and Sherman, 2002*).

Retinogeniculate bouton clustering depends on visual experience, and dark-rearing after eye-opening reduces clustering within individual axon arbors (*Hong et al., 2014*). However, less is known about synaptic spatial relationships and activity-dependent release site refinement during eye-specific competition before eye-opening. During early axon ingrowth, individual RGCs extend sparse side branches in the incorrect eye-specific territory (*Sretavan and Shatz, 1984*; *Sretavan and Shatz, 1986*; *Dhande et al., 2011*). These transient branches form synapses that contain fewer presynaptic vesicles than those made by the same axon in the correct eye-specific domain (*Campbell and Shatz, 1992*). Functional recordings after eye-opening show that while many RGC inputs are pruned, the remaining inputs strengthen by adding release sites (*Chen and Regehr, 2000*). This leads to glutamate spill-over and cross-talk between RGC inputs that increases postsynaptic relay neuron excitability (*Hauser et al., 2014*). Thus, early eye-specific differences in release site number or spacing may contribute to binocular input refinement prior to eye-opening.

In a previous study from our laboratory, we used volumetric stochastic optical reconstruction microscopy (STORM), anterograde tract tracing, and immunohistochemical labeling of synaptic proteins to show that dominant-eye synapses contain more total vesicles and have more vesicles near the AZ (putative docking) compared with non-dominant-eye synapses (*Zhang et al., 2023*). However, we did not examine whether eye-specific inputs differ in AZ number or AZ spatial proximity (clustering) during synaptic competition. To address this gap, we reanalyzed our published dataset, focusing on eye-specific AZ spatial relationships. We found that projections from each eye form a subset of retinogeniculate inputs in which several (2–4) AZs share a common cluster of presynaptic vesicles. We refer to these synapses as multi-active-zone (mAZ) inputs. All other retinogeniculate inputs contained one single active zone and its associated vesicle cluster, and we term these single-active-zone (sAZ) synapses.

During eye-specific synaptic competition, the dominant-eye projection formed more mAZ inputs, each with more AZs and a larger presynaptic vesicle pool compared to the non-dominant-eye projection. Similarly, the dominant eye had higher vesicle signal at sAZ inputs. At the peak of synaptic competition midway through the first postnatal week (postnatal day 4), the non-dominant-eye formed numerous sAZ inputs, equalizing the global synapse density between the two eyes. These eye-specific AZ patterns were disrupted in a mutant mouse line with abnormal stage II cholinergic retinal waves and retinogeniculate segregation defects.

## Results

### Retinogeniculate inputs form multiple active zones during eye-specific competition

To investigate active zone refinement during eye-specific segregation, we reanalyzed a volumetric super-resolution imaging dataset previously published by our laboratory (*Zhang et al., 2023*). We used volumetric STORM (*Vatan et al., 2021*) to image the dLGNs of wild-type (WT) mice at three postnatal ages (P2, P4, and P8) (*Figure 1A*). We labeled eye-specific inputs by monocular injection of Alexa Fluor-conjugated cholera toxin subunit B tracer (CTB) together with immunostaining for presynaptic Bassoon, postsynaptic Homer1, and presynaptic vesicular glutamate transporter 2 (VGluT2) proteins (*Figure 1B*). We collected separate image volumes (~45 K µm$^3$ each) from three biological replicates

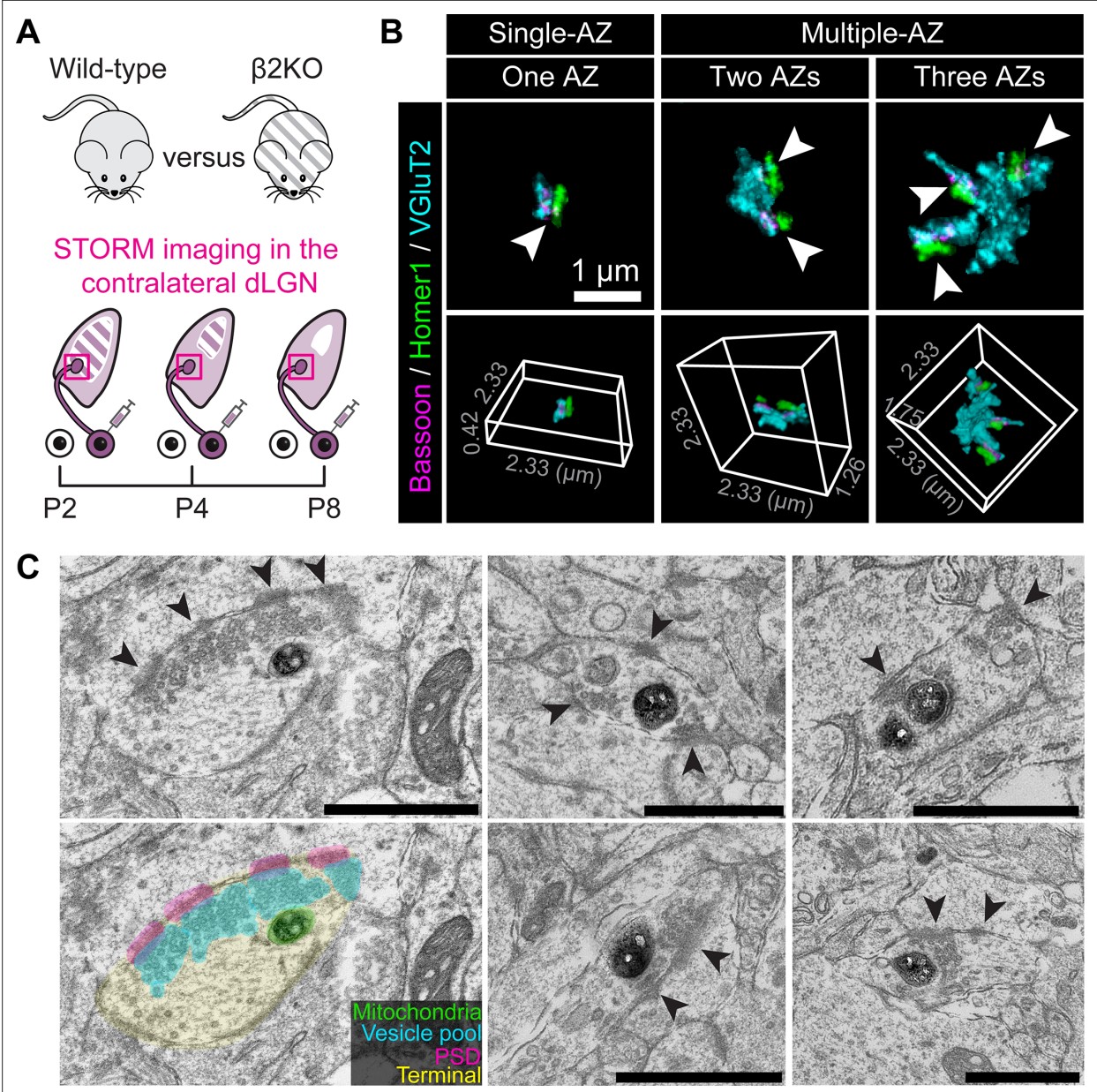

**Figure 1.** Retinogeniculate boutons form multiple active zones (mAZ) during eye-specific competition. (**A**) Experimental design. CTB-Alexa 488 was injected into the right eye of wild-type and β2KO mice. One day after the treatment, tissue was collected from the left dorsal lateral geniculate nucleus (dLGN) at P2, P4, and P8. Red squares indicate the stochastic optical reconstruction microscopy (STORM) imaging regions that were analyzed. (**B**) Representative examples of individual single-active-zone (sAZ) and mAZ inputs, with corresponding active zone counts ranging from one to three. Upper panels show Z-projections of inputs and lower panels show the corresponding 3D volume. Arrowheads point to individual Bassoon clusters (active zones) paired with postsynaptic Homer1 labels within each input. All examples are from a WT P8 sample. (**C**) Electron micrographs of mAZ retinogeniculate inputs in a P8 *SLC6A4^Cre::ROSA26^LSL-Matrix-dAPEX2* mouse. Darkly stained dAPEX2(+) mitochondria are present within ipsilaterally projecting retinal ganglion cell (RGC) terminals. Arrowheads point to electron-dense material at the postsynaptic density, apposed to individual active zones with clustered presynaptic synaptic vesicles.

at each age. To assess the impact of spontaneous retinal activity on synaptic development across the same time period, we performed identical experiments in β2-knockout (β2KO) mice lacking the beta 2 subunit of the nicotinic acetylcholine receptor, a mutation that disrupts spontaneous cholinergic retinal wave activity, eye-specific segregation, and retinogeniculate synapse development (*Dhande et al., 2011*; *Zhang et al., 2023*; *Muir-Robinson et al., 2002*; *Xu et al., 2015*; *Xu et al., 2011*; *Xu et al., 2016*; *Rossi et al., 2001*; *Grubb et al., 2003*; *Sun et al., 2008*; *Stafford et al., 2009*; *Bansal*

*et al., 2000*; *Burbridge et al., 2014*; *Figure 1A*). Because eye-specific segregation is incomplete until ~P8, we limited our re-analysis to the future contralateral eye-specific region of the dLGN, which is reliably identified across all stages of postnatal development (*Figure 1A*, see also Materials and methods).

By analyzing synapses in the contralateral dLGN from 18 mice across three ages and two geno-types (*Supplementary file 1*), STORM revealed two classes of retinogeniculate inputs distinguished by active zone (AZ) number (*Figure 1B*). We defined each retinogeniculate input as a single contig-uous VGluT2 cluster together with all its associated presynaptic (Bassoon) and postsynaptic (Homer1) paired synaptic labels. Using this definition, inputs that had multiple (2–4) Bassoon AZs were classified as mAZ inputs, while those with a single Bassoon AZ were designated sAZ inputs (*Figure 1B*). Most mAZ inputs contained two AZs (~70–90%, varying with age, genotype, and eye-of-origin); smaller proportions contained three AZs (~10–20%) or four or more AZs (<5%) (*Supplementary file 1*).

Each mAZ input could be a single terminal bouton with several AZs, or a cluster of sAZ synapses within separate boutons (*Bickford et al., 2010*; *Monavarfeshani et al., 2018*; *Hammer et al., 2015*; *Morgan et al., 2016*; *Hammer et al., 2014*). To address this, we used electron micros-copy (EM) to image retinogeniculate terminals in the dLGN at P8. We generated a transgenic line expressing mitochondrial matrix-targeted dimeric dAPEX2 reporter (*Zhang et al., 2019*) in ipsi-laterally projecting RGCs (*Koch et al., 2011*; *Su et al., 2024*; *Johnson et al., 2021*), providing unambiguous mitochondrial labeling in ipsiRGC axons. EM images confirmed the presence of individual retinogeniculate boutons with multiple active zones, consistent with our STORM data (*Figure 1C*). Previous EM reconstructions of retinogeniculate inputs reported no evidence of RGC bouton convergence at the end of the first postnatal week (*Monavarfeshani et al., 2018*). Together, these results suggest that mAZ inputs in STORM images are single RGC terminals that house several closely spaced release sites. Hereafter, we use the word 'synapse' only to refer to each partnered active zone (Bassoon/Homer1 pairs); mAZ inputs contain several synapses, while each sAZ input is one synapse.

## Changes in eye-specific input density during synaptic competition

In our previous analysis, we reported global eye-specific synapse densities that reflected the combined mAZ and sAZ inputs. Dominant-eye synapse density was greater than that of the non-dominant eye at P2 and P8, but eye-specific synapse densities were equivalent at P4 during the peak of synaptic competition (*Zhang et al., 2023*). To determine how these two input classes evolve during compe-tition, we quantified the densities and percentages of mAZ and sAZ inputs in WT and β2KO mice (*Figure 2A*). Eye-of-origin for each retinogeniculate input was assigned by colocalizing CTB signal with VGluT2 (*Figure 2B*). Binocular CTB control injections showed that anterograde tracing labeled >97% of retinogeniculate synapses at P4 and P8, ensuring accurate eye-specific assignment (*Zhang et al., 2023*). Within the contralateral eye-specific region of the dLGN, CTB(+) VGluT2 clusters were classi-fied as 'dominant-eye' inputs and CTB(−) VGluT2 clusters as 'non-dominant-eye' inputs (*Figure 2B*).

After false discovery rate (FDR) correction for multiple comparisons within each age and geno-type (see Quantification and statistical analysis; p(adj) values shown in all figures and *Supplemen-tary file 2*), we found that the density of CTB(+) mAZ inputs tended to be higher than CTB(−) mAZ inputs in WT mice (*Figure 2C*, top left). Consistent with this trend, the mAZ input fraction (% of all inputs) was significantly higher for CTB(+) dominant-eye inputs at P4 and P8 (*Figure 2—figure supplement 1*, top). β2KO mice also developed differences in mAZ input density favoring the domi-nant-eye (*Figure 2C*, bottom left; *Figure 2—figure supplement 1*, bottom). For WT sAZ synapses, the dominant eye had a significantly higher synapse density at P2 [p(adj) = 0.026, Cohen's $d$ = 5.31] and trended higher at P8 [p(adj) = 0.072, Cohen's $d$ = 2.73]. At P4, however, sAZ synapse density was equivalent between the eyes (*Figure 2C*, top right; *Supplementary files 2 and 3*). This pattern resulted from non-dominant-eye sAZ synapse addition, which equalized the global input density between the eyes at P4 as we reported previously (5/95% confidence interval, −0.014 to 0.011 synapses/μm$^3$). In β2KO mice at P4, the non-dominant-eye formed fewer sAZ synapses (*Figure 2C*, bottom right; see *Supplementary files 2 and 3*). Thus, β2KO mice with disrupted retinal activity maintain a higher mAZ input fraction in the dominant-eye projection, but sAZ synapse addition is reduced at the peak of competition.

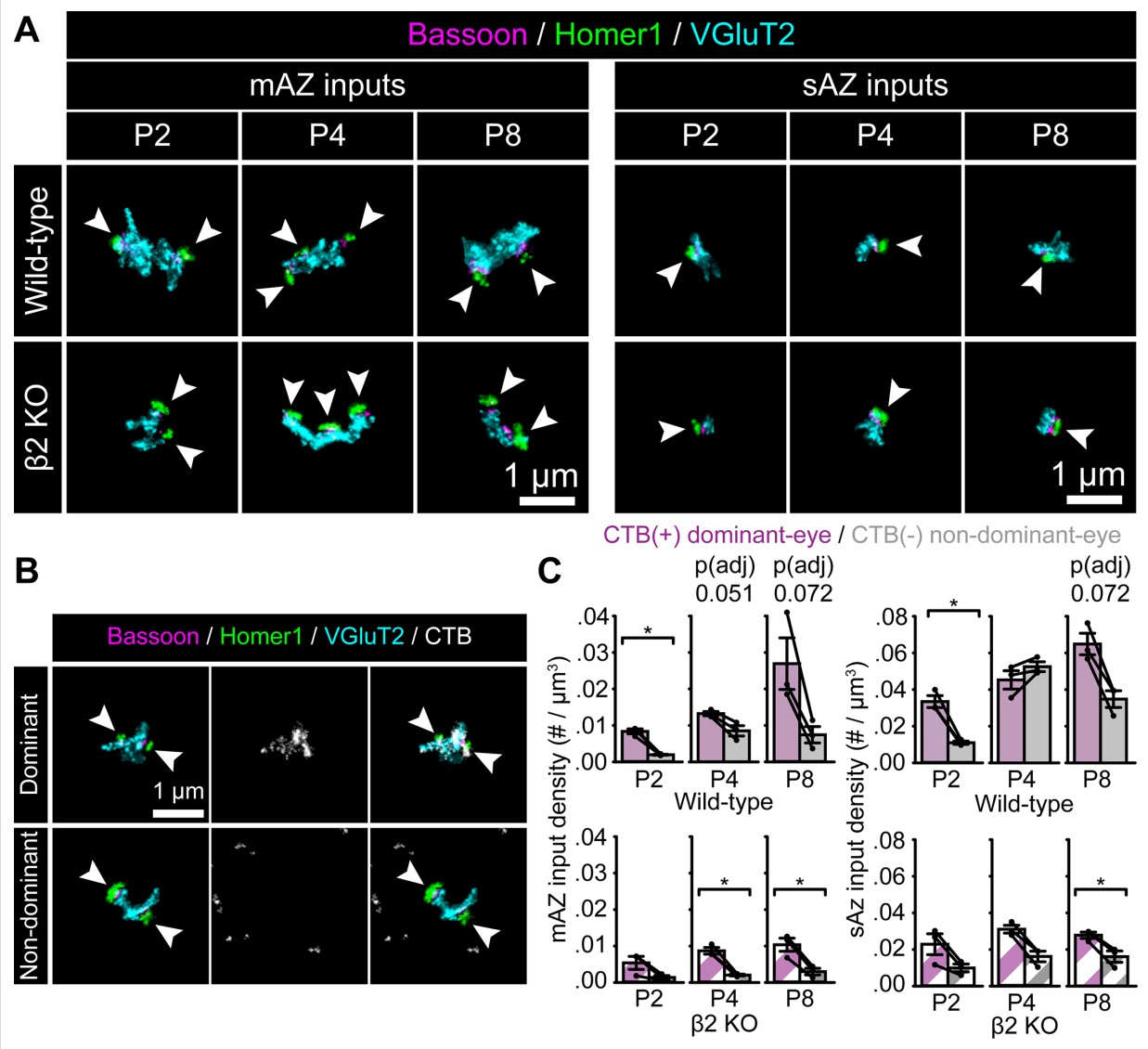

**Figure 2.** Changes in eye-specific input density during synaptic competition. (**A**) Representative Z-projection images of multi-active-zone (mAZ) and single-active-zone (sAZ) inputs across ages and genotypes. Arrowheads point to individual Bassoon/Homer1 cluster pairs indicating release sites. (**B**) Representative CTB(+) dominant-eye (top panels) and CTB(−) non-dominant-eye (bottom panels) mAZ inputs in a WT P8 sample, showing synaptic (left panels), CTB (middle panels), and merged labels (right panels). Arrowheads point to individual Bassoon/Homer1 paired clusters. (**C**) Eye-specific mAZ (left) and sAZ (right) input density across development in WT (top panels) and β2KO mice (bottom panels). Black dots represent mean values from separate biological replicates and black lines connect eye-specific measurements within each replicate ($N = 3$ for each age and genotype). Error bars represent group means ± SEMs. Statistical significance between eye-specific measurements was assessed for each genotype using two-tailed paired *T*-tests with Benjamini–Hochberg false discovery rate (FDR) correction ($\alpha = 0.05$) at each age. *$p(adj) < 0.05$.

The online version of this article includes the following figure supplement(s) for figure 2:

**Figure supplement 1.** Fraction of multi-active-zone (mAZ) inputs across development, related to *Figure 2*.

## mAZ and sAZ inputs from the dominant eye show increased vesicle pool size and vesicle proximity to the active zone

We previously reported a dominant-eye bias in VGluT2 volume when considering all retinogeniculate inputs (*Zhang et al., 2023*). Here, we assessed presynaptic maturation separately in sAZ and mAZ inputs by measuring their total VGluT2 volume. In WT mice, both mAZ (*Figure 3A*, left) and sAZ (*Figure 3B*, left) inputs showed significant eye-specific volume differences in the middle of eye-specific competition at P4. At this age in WT mice, the median VGluT2 cluster volume in dominant-eye mAZ inputs was ~3.55 ± 1.3 μm³ larger (mean ± SE) than that of non-dominant-eye inputs (*Figure 3A*,

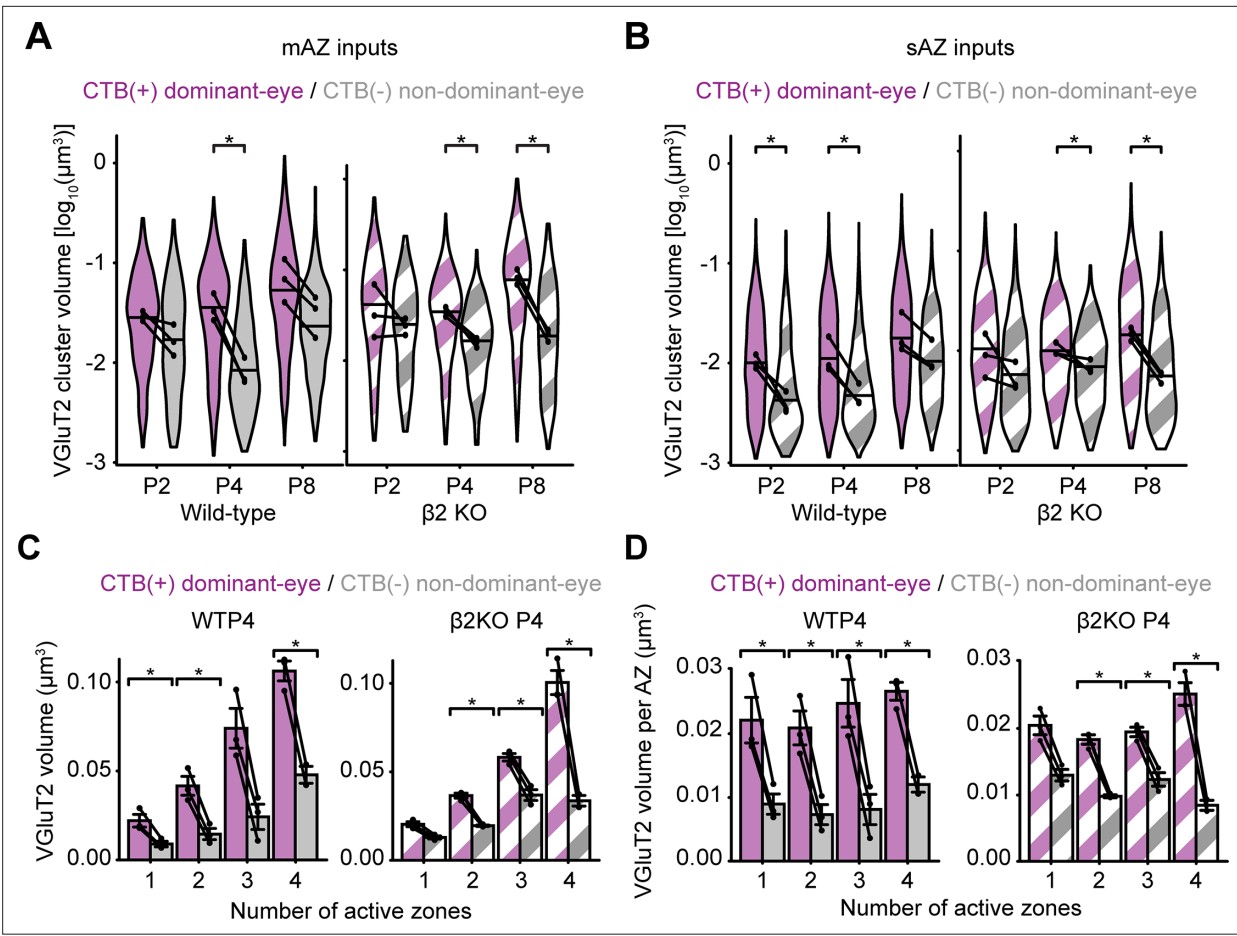

**Figure 3.** Dominant-eye inputs show larger vesicle pools that scale with active zone number. Violin plots showing the distribution of VGluT2 cluster volume for (**A**) multi-active-zone (mAZ) and (**B**) single-active-zone (sAZ) inputs in WT (filled) and β2KO mice (striped) at each age. The width of each violin plot reflects the relative synapse proportions across the entire grouped dataset at each age ($N = 3$ biological replicates) and the maximum width was normalized across all groups. The black dots represent the median value of each biological replicate ($N = 3$), and the black horizontal lines represent the median value of all inputs grouped across replicates. Black lines connect measurements of CTB(+) and CTB(−) populations from the same biological replicate. Statistical significance was determined using a linear mixed model ANOVA with a post hoc Bonferroni correction, followed by Benjamini–Hochberg false discovery rate (FDR) correction ($\alpha = 0.05$) for multiple comparisons at each age/genotype. Black asterisks indicate significant eye-specific differences at each age. *p(adj) < 0.05. (**C**) Eye-specific VGluT2 signal volume for all inputs separated by number of AZs in WT (left panel) and β2KO mice (right panel) at P4. (**D**) Average VGluT2 volume per AZ for all inputs separated by number of AZs in WT (left panel) and β2KO mice (right panel) at P4. In panels (**C**) and (**D**), error bars indicate group means ± SEMs ($N = 3$ biological replicates for each age and genotype). Black dots represent mean values from separate biological replicates and black lines connect eye-specific measurements within each replicate. Statistical significance between eye-specific measurements was assessed for each genotype using two-tailed paired $T$-tests with Benjamini–Hochberg FDR correction ($\alpha = 0.05$): *p(adj) < 0.05.

The online version of this article includes the following figure supplement(s) for figure 3:

**Figure supplement 1.** Quantification of docked vesicle pool volume and AZ number in multi-active-zone (mAZ) and single-active-zone (sAZ) inputs, related to *Figure 3*.

**Figure supplement 2.** Relationship between vesicle pool volume and active zone number, related to *Figure 3*.

left). In contrast, β2KO mice showed a smaller ~$1.9 \pm 1.1$ µm³ (mean ± SE) volume difference between median eye-specific mAZ inputs at the same age (*Figure 3A*, right panel). For sAZ synapses at P4, the magnitudes of eye-specific differences in median VGluT2 volume (mean ± SE) were ~$2.1 \pm 1.0$ µm³ in WT (*Figure 3B*, left) and ~$1.5 \pm 1.1$ in β2KO mice (*Figure 3B*, right). Thus, both mAZ and sAZ vesicle pool volumes are larger for the dominant eye, with the largest eye-specific differences seen for mAZ inputs in WT mice (see *Supplementary file 3*).

In addition to total vesicle pool volume, we quantified the readily releasable pool by measuring VGluT2 volume within a 70-nm shell around each AZ, considering this a proxy for docked vesicles

(*Figure 3—figure supplement 1A–C*; *Zhang et al., 2023*). In WT mice at P4, dominant-eye inputs showed greater vesicle volume per AZ than non-dominant inputs, in both mAZ and sAZ terminals (*Figure 3—figure supplement 1B*, left; *Supplementary file 3*). These eye-specific differences were absent in β2KO mice (*Figure 3—figure supplement 1B*, right; *Supplementary file 3*). However, when comparing mAZ and sAZ inputs from the same eye, vesicle volume per AZ was similar across all ages and genotypes (*Figure 3—figure supplement 1A–C*; *Supplementary file 2*). This confirms our previous finding that vesicle docking favors the dominant eye (*Zhang et al., 2023*) and shows that AZs formed by a single eye have similar docking levels in both their mAZ and sAZ terminals.

## Vesicle pool size scales with active zone number

Because mAZ inputs showed greater total VGluT2 volume than sAZ synapses, yet exhibited comparable vesicle docking, the disparity could reflect a scaling effect of vesicle pool size with increased AZ number. To evaluate how presynaptic vesicle pool volume scales with AZ number, we compared the Bassoon cluster number to VGluT2 volume for every retinogeniculate input. In both WT (*Figure 3—figure supplement 1D*) and β2KO mice (*Figure 3—figure supplement 1E*), mAZ inputs contained an average of two to three Bassoon clusters (separate AZs) in the first postnatal week. In WT mice, CTB(+) dominant-eye mAZ inputs contained more AZs than CTB(−) non-dominant-eye mAZ inputs at P4 and P8 (*Figure 3—figure supplement 1D*). This maturation was delayed until P8 in β2KO mice (*Figure 3—figure supplement 1E*). For CTB(+) dominant-eye inputs in both genotypes at P4, vesicle pool volume correlated positively with AZ number (*Figure 3C*; Pearson correlation coefficients: WT [0.99] and β2KO [0.95]). A similar, but weaker correlation was observed for CTB(−) non-dominant-eye inputs (Pearson correlation coefficients: WT [0.97] and β2KO [0.90]). Dividing the total presynaptic VGluT2 volume by the AZ number revealed a consistent vesicle volume per AZ for both sAZ and mAZ inputs (*Figure 3D*; *Figure 3—figure supplement 2*; *Supplementary file 3*). Collectively, these results indicate that presynaptic vesicle pool volume scales with AZ number for each eye-specific input, while a dominant-eye bias persists throughout development.

## Synapse clustering before eye-opening

Eye-specific competition is thought to involve stabilization of coactive, neighboring inputs from the same eye and elimination of out-of-sync inputs from the opposite eye (*Assali et al., 2014*; *Fassier and Nicol, 2021*; *Penn et al., 1998*). Because axon refinement depends on the relative strength of neuro-transmission between competing inputs (*Koch et al., 2011*; *Assali et al., 2017*; *Fredj et al., 2010*; *Hua et al., 2005*; *Rahman et al., 2020*; *Munz et al., 2014*; *Zhang et al., 1998*; *Matsumoto et al., 2024*), mAZ inputs with multiple release sites could help stabilize nearby like-eye inputs (*Rahman et al., 2020*; *Yasuda et al., 2021*; *Louail et al., 2020*; *Kutsarova et al., 2023*).

To quantify synaptic clustering patterns, we measured the distance from every eye-specific sAZ synapse to all other sAZ and mAZ inputs within each image field. sAZ synapses were often found nearby other inputs from the same eye (*Figure 4A*). To define clustering, we searched volumetrically around each mAZ input and measured the fraction of sAZ synapses that were nearby at increasing distances (*Figure 4—figure supplement 1A, B*). We then compared the observed percentages with datasets in which sAZ synapse positions were randomly shuffled within each neuropil volume. The largest differences occurred within a 1- to 2-µm search distance (*Figure 4—figure supplement 1A, B*), and so we chose a 1.5-µm cutoff from the edge of each mAZ input to designate it 'clustered' if at least one neighboring sAZ synapse lay within this distance and 'isolated' if none did (*Figure 4B*). No significant clustering was detected when sAZ and mAZ inputs originated from opposite eyes (*Figure 4—figure supplement 1C, D*).

At the peak of competition in WT mice (P4), more than 65% of both mAZ and sAZ inputs were clustered for both eyes, while this proportion fell to ~50% in β2KO mice (*Figure 4B*). In the WT dominant-eye projection, the fractions of clustered mAZ and sAZ inputs were similar at each age (*Figure 4C*, top left panel). For the non-dominant-eye projection, however, there were slightly more clustered mAZ inputs compared to clustered sAZ inputs at P4 (*Figure 4C*, bottom left panel), the age when this eye adds sAZ synapses (*Figure 2C*). β2KO mice showed no difference between mAZ and sAZ clustering at any age (*Figure 4C*, right panels). Additionally, in WT mice at P4 and P8, clustered mAZ inputs from both eyes had marginally more neighboring sAZ synapses than did clustered sAZ synapses (*Figure 4D*, left); this enrichment was absent in β2KO mice (*Figure 4D*, right). Thus, while

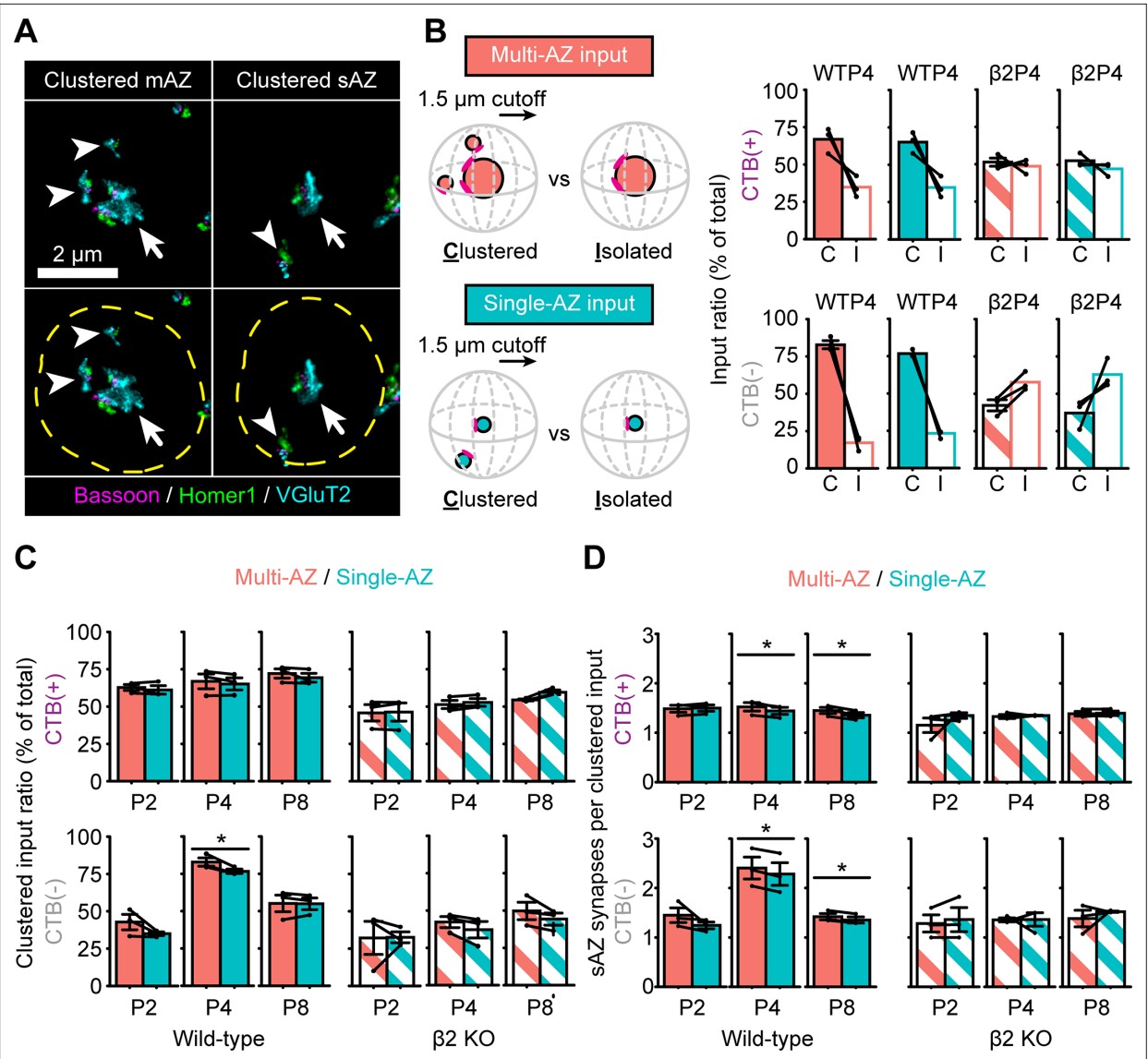

**Figure 4.** Eye-specific synapse clustering before eye-opening. (**A**) Representative multi-active-zone (mAZ, left panels) and single-active-zone (sAZ, right panels) inputs in a WT P8 sample with nearby sAZ synapses (arrowheads) clustered within 1.5 µm (dashed yellow ring). Arrows point to the centered mAZ or sAZ inputs. (**B**) Ratio of clustered and isolated mAZ and sAZ inputs for CTB(+) (upper panels) and CTB(−) (lower panels) inputs in WT and β2KO mice at P4. (**C**) Comparison of the clustered input ratio between mAZ and sAZ inputs across different ages, genotypes, and eyes of origin. (**D**) Comparison of the average number of nearby sAZ synapses for clustered mAZ and sAZ inputs across different ages, genotypes, and eyes of origin. In panels B–D, black dots represent mean values from separate biological replicates and black lines connect measurements within each replicate (N = 3 for each age and genotype). Error bars represent group means ± SEMs. For each genotype, two-tailed paired *T*-tests with Benjamini–Hochberg false discovery rate (FDR) correction (α = 0.05) were used to test statistical significance between mAZ and sAZ inputs at each age. *p(adj) < 0.05.

The online version of this article includes the following figure supplement(s) for figure 4:

**Figure supplement 1.** Single-active-zone (sAZ) synapse clustering near like-eye multi-active-zone (mAZ) inputs, related to *Figure 4*.

most retinogeniculate synapses lie within 1.5 µm of another like-eye input, WT mice show a tendency toward forming more synapses near mAZ inputs during synaptic competition.

Clustered mAZ inputs in P4 WT mice were also closer together than isolated mAZ inputs. Clustered mAZ inputs formed by the dominant eye were ~32% closer to the nearest like-eye clustered mAZ input compared to isolated mAZ inputs (*Figure 5A*, left panel). In the non-dominant-eye projection, clustered mAZ inputs were ~55% closer together compared to isolated mAZ inputs (*Figure 5B*, left panel). Once segregation was complete at P8, distances between clustered and isolated mAZ inputs were more similar (*Figure 5—figure supplement 1*). In β2KO mice, distances between isolated and

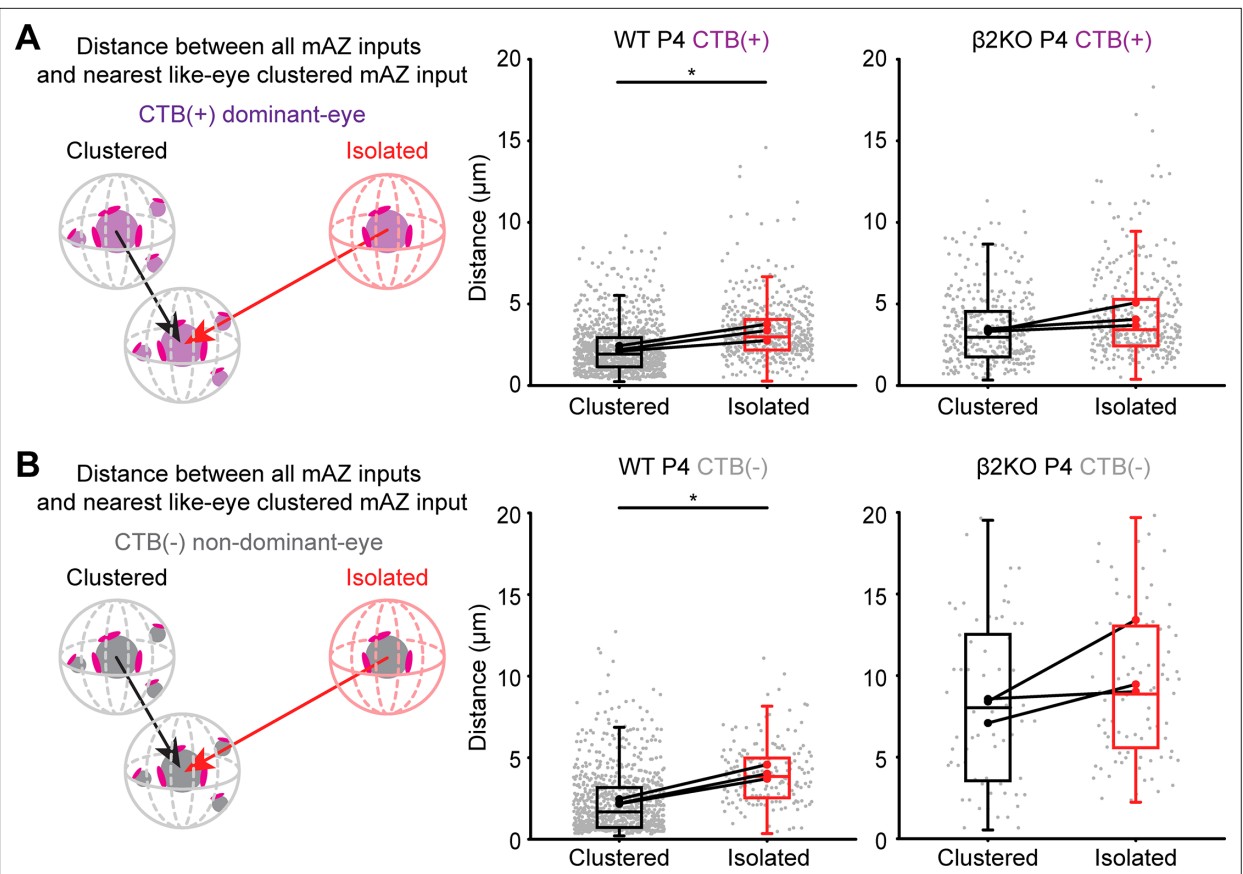

**Figure 5.** Clustered multi-active-zone (mAZ) inputs are closer than isolated inputs during competition. Distance between clustered and isolated mAZ inputs and the closest like-eye clustered mAZ input, shown for (**A**) CTB(+) and (**B**) CTB(−) projections at P4 in WT and β2KO mice. Boxes indicate the 25–75% distribution of input measurements from $N = 3$ biological replicates, and whiskers extend to 1.5 times the interquartile range. Gray dots represent individual distance measurements for all mAZ inputs. Black and red dots represent mean values from separate biological replicates, and black lines connect measurements within each replicate ($N = 3$ for each age and genotype). Statistical significance was determined using a linear mixed model ANOVA with post hoc Bonferroni correction. For each genotype, p-values were corrected for multiple testing with Benjamini–Hochberg false discovery rate (FDR) correction ($\alpha = 0.05$) at each age. Black asterisks indicate significant differences. *p(adj) < 0.05.

The online version of this article includes the following figure supplement(s) for figure 5:

**Figure supplement 1.** Clustered and isolated multi-active-zone (mAZ) inputs show similar spacing after competition, related to **Figure 5**.

**Figure supplement 2.** Single-active-zone (sAZ) synapse vesicle pool volume is independent of distance to multi-active-zone (mAZ) inputs, related to **Figure 5**.

clustered mAZ inputs and their nearest clustered mAZ neighbor did not differ for either eye at P4 or P8 (**Figure 5**, right panels; **Figure 5—figure supplement 1**, right panels). These patterns are consistent with sAZ synapse addition when mAZ inputs are in close proximity during competition.

Finally, we tested whether sAZ vesicle pool volume varies with distance to mAZ inputs. We classified each sAZ as 'near' (≤1.5 µm) or 'far' (>1.5 µm) relative to the nearest like-eye mAZ input. Across ages and genotypes, vesicle pool volume was similar for near and far sAZ synapses in both eyes (**Figure 5—figure supplement 2**). Thus, vesicle pool size in sAZ synapses appears independent of their proximity to mAZ inputs.

## Discussion

Spontaneous retinal activity guides eye-specific refinement by strengthening dominant eye inputs in the correct territory and pruning weaker inputs from the competing eye. Using volumetric super-resolution microscopy and eye-specific synaptic immunolabeling, we previously identified activity-dependent, eye-specific differences in presynaptic vesicle pool size and vesicle association with the

active zone (AZ), both favoring the dominant eye during synaptic competition (*Zhang et al., 2023*). Here, we reanalyzed this dataset to measure synaptic spatial arrangements during normal development and how these are affected in a mutant model with abnormal eye-specific segregation.

## Early input maturation and addition of release sites

Before eye-opening, RGC axons from both eyes formed terminals with either multiple active zones (mAZ inputs) or a single active zone (sAZ synapses). EM of dAPEX2-labeled ipsilateral RGC inputs revealed individual boutons with multiple active zones. Prior EM studies in mice showed no clustering from neighboring RGC boutons at postnatal day 8 (*Bickford et al., 2010*; *Monavarfeshani et al., 2018*), suggesting the mAZ inputs seen in STORM images are single boutons containing multiple release sites. During competition, the dominant eye projection generated more of these mAZ inputs, each with more active zones and larger vesicle pools than the non-dominant eye. We observed a linear relationship between presynaptic vesicle signal volume and active zone number, suggesting that release site addition and vesicle pool expansion may be linked during input maturation. Functional recordings indicate that release site addition is the primary driver of retinogeniculate input strengthening after eye-opening (*Chen and Regehr, 2000*). Our findings reveal that eye-specific differences in release site addition at individual terminals emerge at the earliest stages of binocular refinement.

## Spatial interactions during competition

Ipsilateral RGC axons are delayed in entering the dLGN until after contralateral inputs have already innervated the nucleus (*Godement et al., 1984*). Despite their late arrival, these axons competed by forming numerous sAZ inputs, which equalized the overall input density between the eyes in the future contralateral eye domain at postnatal day 4 (P4). Most of these synapses formed nearby other like-eye neighbors and were enriched near mAZ inputs, supporting the idea of release site addition at strong synapses. Neighboring sAZ synapses could ultimately mature into mAZ inputs as release sites are added and vesicle pools expand.

Distance analysis showed that clustered mAZ boutons were closer together than isolated mAZ boutons, further supporting release site addition near strong neighboring inputs. It remains unknown, however, whether nearby mAZ inputs originate from the same RGC. Nearby mAZ inputs within individual RGC arbors could promote release site addition via intrinsic mechanisms that scale with release site number, such as increased presynaptic calcium entry or surface delivery of synaptogenic molecules via presynaptic release. Boutons with multiple release sites could also trigger non-cell-autonomous signaling mechanisms that stabilize neighboring inputs from the same eye (*Rahman et al., 2020*; *Louail et al., 2020*; *Kutsarova et al., 2023*). Release site addition increases extracellular glutamate concentration and promotes spillover onto adjacent synapses, which enhances the excitability of developing relay cells (*Hauser et al., 2014*) and could contribute to long-term synaptic plasticity evoked by high-frequency spiking in retinal wave bursts (*Mooney et al., 1993*; *Ziburkus et al., 2009*; *Butts et al., 2007*; *Lee et al., 2014*). Consequently, release site addition and local bouton clustering may act in tandem to stabilize coactive inputs.

Competitive refinement also involves synapse elimination and axonal retraction through punishment signals. Genetic deletions of VGluT2 or RIM1 proteins in ipsilaterally projecting RGCs decreased presynaptic vesicle release and prevented retraction of contralateral RGC axons from the ipsilateral territory (*Koch et al., 2011*; *Assali et al., 2017*). One downstream mediator of synaptic punishment is JAK2 kinase, which is phosphorylated in less active synapses (*Yasuda et al., 2021*). Similar to neurotransmission mutant phenotypes, disruption of JAK2 signaling prevents axon retraction (punishment) during competition (*Yasuda et al., 2021*). The spatial analyses we developed here will enable future mapping of input-specific punishment signals during synaptic competition. This includes phospho-JAK2 as well as molecular tags for glial pruning of weak inputs during eye-specific segregation (*Chung et al., 2013*; *Stevens et al., 2007*; *Schafer et al., 2012*).

## Requirement for spontaneous retinal activity

β2KO mice showed significant defects in synapse addition during the first postnatal week. mAZs still formed with similar overall ratios as in WT controls (*Figure 2—figure supplement 1*; *Supplementary file 3*) and eye-specific differences in vesicle pool size still emerged. However, β2KO mice failed to form many sAZ synapses at the height of competition (P4), particularly in the late-arriving

non-dominant (ipsilateral)-eye projection. The failure to add synapses could explain the observation that synaptic clustering was reduced and more inputs formed in isolation in the mutants compared to controls.

While our results highlight developmental changes in presynaptic release site addition and clustering, activity-dependent postsynaptic mechanisms also influence input refinement at later stages. Retinogeniculate synapses undergo postsynaptic strengthening and weakening through potentiation and depression mediated by AMPARs and NMDARs (*Mooney et al., 1993*; *Ziburkus et al., 2009*; *Butts et al., 2007*; *Lee et al., 2014*). After eye-specific segregation, spontaneous retinal activity is required for postsynaptic AMPAR insertion, synaptic strengthening, and elimination of weaker inputs (*Hooks and Chen, 2006*). Continued maintenance of segregation depends on calcium influx into relay neurons via L-type calcium channels, further implicating postsynaptic signaling in late-stage refinement (*Dilger et al., 2015*; *Cork et al., 2001*). Input targeting may also be guided by molecular cues to form non-random, eye-specific connections with postsynaptic targets. Ipsilaterally projecting RGCs have distinct gene expression profiles that specify axon guidance and may further support eye-specific synaptic targeting (*Fernández-Nogales et al., 2022*; *Wang et al., 2016*). Spontaneous retinal activity may permit axons to read out molecular regulators of synaptogenesis, as previously shown for RGC axon retraction (*Nicol et al., 2007*).

### Release site addition as a general mechanism underlying synaptic competition

Early synaptic clustering during retinogeniculate development resembles other circuits where neural activity guides competitive synaptic and axonal remodeling. At the neuromuscular junction (NMJ), motor neuron terminals compete for control of a postsynaptic muscle fiber; a single motor axon input strengthens while competing axons are eliminated (*Balice-Gordon et al., 1993*; *Gan and Lichtman, 1998*; *Wyatt and Balice-Gordon, 2003*). Competition at the NMJ depends on inter-synaptic distance, with motor axons losing connections located near stronger competing synapses (*Balice-Gordon et al., 1993*; *Gan and Lichtman, 1998*). As winning terminals are enlarged, presynaptic release site addition maintains a consistent density of Bassoon clusters (~2–3/μm$^2$) (*Chen et al., 2012*). Neighboring inputs with weaker synaptic transmission are selectively eliminated (*Balice-Gordon and Lichtman, 1994*; *Buffelli et al., 2003*). Competing motor axons differ in their release probabilities early in development (*Kopp et al., 2000*), suggesting that presynaptic efficacy triggers local 'stabilization' signals in winners and 'punishment' signals in losers (*Sanes and Lichtman, 1999*). Presynaptic agrin release stabilizes postsynaptic receptor clusters by counteracting the destabilizing, dispersal effects of acetylcholine release, emphasizing the importance of early presynaptic transmission in postsynaptic stabilization (*Misgeld et al., 2005*; *Lin et al., 2005*).

Similarly, in the developing cerebellum, Purkinje cells initially receive inputs from multiple climbing fibers (CFs) during the first postnatal week. Subsequently, one winning CF emerges and consolidates its synaptic inputs, while losing CFs are pruned (*Hashimoto and Kano, 2013*; *Bosman and Konnerth, 2009*). Here again, the winning input strengthens by adding presynaptic release sites, which increases multivesicular release and elevates glutamate concentration in the synaptic cleft (*Hashimoto and Kano, 2003*; *Nitta et al., 2025*; *Wilson et al., 2019*). Postsynaptic ultrastructural and molecular changes occur several days later as dominant CF inputs potentiate and losing CF inputs depress (*Nitta et al., 2025*; *Bosman et al., 2008*; *Ohtsuki and Hirano, 2008*) through spike-timing-dependent remodeling (*Lorenzetto et al., 2009*; *Kawamura et al., 2013*). Across these models, the earliest distinguishing feature between competing inputs is relative presynaptic transmission strength. Thus, release-site addition may be a conserved mechanism that biases synaptic refinement outcomes across developing neural circuits.

## Materials and methods

### Key resources table

| Reagent type (species) or resource | Designation | Source or reference | Identifiers | Additional information |
|---|---|---|---|---|
| Genetic reagent (*Mus musculus*, male/female) | C57BL/6J; wild-type; WT | The Jackson Laboratory | RRID:IMSR_JAX:000664 | Ages P2–P8 |

*Continued on next page*

*Continued*

| Reagent type (species) or resource | Designation | Source or reference | Identifiers | Additional information |
|---|---|---|---|---|
| Genetic reagent (*Mus musculus*, male/female) | β2-nAChR$^{-/-}$; *CHRNB2* KO; β2KO | PMC4258148 | | Ages P2–P8 |
| Genetic reagent (*Mus musculus*, male/female) | *Tg(Slc6a4-cre)ET33Gsat/Mmucd; BAC-Cre Slc6a4-33* | MMRRC | RRID:MMRRC_017260-UCD | Age P8 |
| Genetic reagent (*Mus musculus*, male/female) | *Gt(ROSA)26Sor$^{tm1.1(CAG-COX4I1/APX1*)Ddg}$/J; ROSA26$^{LSL-Matrix-dAPEX2}$* | The Jackson Laboratory | RRID:IMSR_JAX:032765 | Age P8 |
| Antibody | Donkey anti-Guinea pig IgG unconjugated | Jackson ImmunoResearch | Cat# 706-005-148; RRID:AB_2340443 | (1:100) |
| Antibody | Donkey anti-Mouse IgG unconjugated | Jackson ImmunoResearch | Cat# 715-005-150; RRID:AB_2340758 | (1:100) |
| Antibody | Donkey anti-Rabbit IgG unconjugated | Jackson ImmunoResearch | Cat# 711-005-152; RRID:AB_2340585 | (1:100) |
| Antibody | Guinea pig polyclonal anti-VGluT2 | Millipore Sigma | AB2251-I; RRID:AB_2665454 | (1:100) |
| Antibody | Mouse monoclonal anti-Bassoon | Abcam | Ab82958; RRID:AB_1860018 | (1:100) |
| Antibody | Rabbit polyclonal anti-Homer1 | Synaptic Systems | Cat# 160 003; RRID:AB_887730 | (1:100) |
| Sequence-based reagent | *CHRNB2_F* | PMC4258148 | PCR primers | CAGGCGTTATCCACAAAGACAGA |
| Sequence-based reagent | *CHRNB2_R* | PMC4258148 | PCR primers | TTGAGGGGAGCAGAACAGAATC |
| Sequence-based reagent | *CHRNB2_mutant_R* | PMC4258148 | PCR primers | ACTTGGGTTTGGGCGTGTTGAG |
| Sequence-based reagent | *SLC6A4_F* | MMRRC | PCR primers | GGTCCTTGGCAGATGGGCAT |
| Sequence-based reagent | *SLC6A4_R* | MMRRC | PCR primers | CGGCAAACGGACAGAAGCATT |
| Sequence-based reagent | *ROSA26$^{LSL-Matrix-dAPEX2}$_WT_F* | The Jackson Laboratory | PCR primers | CTGGCTTCTGAGGACCG |
| Sequence-based reagent | *ROSA26$^{LSL-Matrix-dAPEX2}$_WT_R* | The Jackson Laboratory | PCR primers | AATCTGTGGGAAGTCTTGTCC |
| Sequence-based reagent | *ROSA26$^{LSL-Matrix-dAPEX2}$_mutant_F* | The Jackson Laboratory | PCR primers | CCATCAGCACCAGCGTGT |
| Sequence-based reagent | *ROSA26$^{LSL-Matrix-dAPEX2}$_mutant_R* | The Jackson Laboratory | PCR primers | GAACCCTTAGTGGGATCGGG |
| Peptide, recombinant protein | Catalase from bovine liver | Sigma-Aldrich | C1345 | |
| Peptide, recombinant protein | Normal donkey serum | Jackson ImmunoResearch | Cat# 017-000-121 | |
| Peptide, recombinant protein | Glucose oxidase | Sigma-Aldrich | G2133 | |
| Commercial assay or kit | EMbed 812 embedding kit with BDMA | Electron Microscopy Sciences | Cat# 14121 | |
| Commercial assay or kit | UltraBed Kit | Electron Microscopy Sciences | Cat# 14310 | |
| Chemical compound, drug | Alexa Fluor 405 NHS-ester | Thermo Fisher Scientific | Cat# A30000 | |
| Chemical compound, drug | Alexa Fluor 647 NHS-ester | Thermo Fisher Scientific | Cat# A20006 | |
| Chemical compound, drug | Atto 488 NHS-ester | ATTO-TEC GmbH | AD 488-31 | |
| Chemical compound, drug | Cacodylic acid- sodium cacodylate, trihydrate | Electron Microscopy Sciences | Cat# 12300 | |
| Chemical compound, drug | Calcium chloride | Electron Microscopy Sciences | Cat# 12340 | |
| Chemical compound, drug | Chloroform | Sigma-Aldrich | 288306 | |
| Chemical compound, drug | Cy-3B mono NHS-ester | Cytiva | PA63101 | |
| Chemical compound, drug | Cysteamine | Sigma-Aldrich | 30070 | |
| Chemical compound, drug | DY-749P1 NHS-ester | Dyomics GmbH | Cat# 749P1-01 | |
| Chemical compound, drug | Dulbecco's phosphate buffered saline | Sigma-Aldrich | D8662 | |
| Chemical compound, drug | Ethanol | Pharmco | Cat# 111000200C1GL | |
| Chemical compound, drug | FluoSpheres Infrared (715/755) | Invitrogen | Cat# F8799 | |

*Continued on next page*

*Continued*

| Reagent type (species) or resource | Designation | Source or reference | Identifiers | Additional information |
|---|---|---|---|---|
| Chemical compound, drug | FluoSpheres Orange (540/560) | Invitrogen | Cat# F8809 | |
| Chemical compound, drug | D-(+)-Glucose | Sigma-Aldrich | G7528 | |
| Chemical compound, drug | DAB (diaminobenzidine) | Sigma-Aldrich | RES2041D | |
| Chemical compound, drug | Glutaraldehyde 70%, EM Grade | Electron Microscopy Sciences | Cat# 16360 | |
| Chemical compound, drug | Glycine | Sigma-Aldrich | G7126 | |
| Chemical compound, drug | Hydrogen peroxide, 30% | Thermo Fisher Scientific | Cat# BP2633500 | |
| Chemical compound, drug | L-Aspartic acid | Fisher Scientific | Cat# A13520 | |
| Chemical compound, drug | Lead nitrate | Electron Microscopy Sciences | Cat# 17900 | |
| Chemical compound, drug | Osmium tetroxide 4% aqueous solution | Electron Microscopy Sciences | Cat# 19140 | |
| Chemical compound, drug | Paraformaldehyde 16%, EM Grade | Electron Microscopy Sciences | Cat# 15710 | |
| Chemical compound, drug | Potassium ferricyanide | Electron Microscopy Sciences | Cat# 20150 | |
| Chemical compound, drug | Propylene oxide | Electron Microscopy Sciences | Cat# 20401 | |
| Chemical compound, drug | Sodium azide | Sigma-Aldrich | S2002 | |
| Chemical compound, drug | Sodium chloride | Sigma-Aldrich | S9888 | |
| Chemical compound, drug | Sodium hydroxide pellets | Sigma-Aldrich | 567530 | |
| Chemical compound, drug | Thiocarbohydrazide | Electron Microscopy Sciences | Cat# 21900 | |
| Chemical compound, drug | Tris-base (Trizma-base) | Sigma-Aldrich | T8524 | |
| Chemical compound, drug | Triton X-100 | Sigma-Aldrich | X100PC | |
| Chemical compound, drug | Uranyl acetate | Electron Microscopy Sciences | Cat# 22400 | |
| Software, algorithm | 3D-DAOSTORM analysis (single-molecule localization fitting code); version 2.1 | PMC:PMC4243665 | | https://github.com/ZhuangLab/storm-analysis |
| Software, algorithm | Fiji (ImageJ) | PMC:PMC3855844 | | https://fiji.sc |
| Software, algorithm | MATLAB | MathWorks | | https://mathworks.com |
| Software, algorithm | Python3 | Python | | https://www.python.org |
| Software, algorithm | Rstudio | Posit | | https://posit.co/ |
| Software, algorithm | SPSS | IBM | | https://www.ibm.com/products/spss-statistics |
| Software, algorithm | STORM acquisition control code (packages include hal4000.py, steve.py, and dave.py); version V2019.06.28 | Zhuang Laboratory, Harvard University | | https://github.com/ZhuangLab/storm-control |
| Other | 5 min epoxy in DevTube | Jenson Tools | Cat# 14250 | |
| Other | BEEM embedding capsules | Electron Microscopy Sciences | Cat# 70020-B | |
| Other | Coverslip No. 1.5 (24 mm × 30 mm) | VWR | Cat# 48404-467 | |
| Other | Custom-built STORM microscope | PMC:PMC8637648 | | Information on our build is available from the Corresponding Author |
| Other | Gilder thin bar hexagonal mesh grids | Electron Microscopy Sciences | Cat# T200H-Cu | |
| Other | Microscope slides | VWR | Cat# 16004-422 | |

The raw imaging data in this paper were previously reported (*Zhang et al., 2023*). Materials and methods below are adapted from this work. All MATLAB and Python code used in the work is available on GitHub (https://github.com/SpeerLab/Aligned_data_analysis_SynapseClustering; copy archived at

*Zhang and Speer, 2025*). Raw STORM images of the full data are available on the open-access Brain Imaging Library (*Benninger et al., 2020*). These images can be accessed here https://doi.org/10.35077/g.1164.

## Animals

WT C57BL/6J mice (Stock Number 000664) and *ROSA26^{LSL-Matrix-dAPEX2}* mice (Stock Number 032765) used in this study are available from The Jackson Laboratory (Bar Harbor, Maine). *SLC6A4^{Cre}* (cre recombinase expression under the serotonin transporter promoter), β2KO (genetic deletion of *CHRNB2* encoding the β2 subunit of the nicotinic acetylcholine receptor), and *ROSA26^{LSL-Matrix-dAPEX2}* (cre-dependent expression of dimeric APEX2 targeted to the mitochondrial matrix) mice were generously gifted by Drs. Eric M. Ullian (University of California, San Francisco), Michael C. Crair (Yale School of Medicine), and Joshua H. Singer (University of Maryland), respectively. All experimental procedures were performed in accordance with an animal study protocol approved by the Institutional Animal Care and Use Committee (IACUC) at the University of Maryland. Neonatal male and female mice were used interchangeably for all experiments. Tissue from biological replicates (N = 3 animals) was collected for each age (P2/P4/P8) from each genotype (WT and β2KO) (18 animals total). Primers used for genotyping β2KO mice are: forward: CAGGCGTTATCCACAAAGACAGA; reverse: TTGAGGGGAGCAGAACAGAATC; mutant reverse: ACTTGGGTTTGGGCGTGTTGAG. Primers used for genotyping *SLC6A4^{Cre}* mice are: forward: GGTCCTTGGCAGATGGGCAT; reverse: CGGCAAAC GGACAGAAGCATT. Primers used for genotyping *ROSA26^{LSL-Matrix-dAPEX2}* mice are: WT forward: CTGG CTTCTGAGGACCG; WT reverse: AATCTGTGGGAAGTCTTGTCC; mutant forward: CCATCAGC ACCAGCGTGT; mutant reverse: GAACCCTTAGTGGGATCGGG.

## EM tissue preparation

Animals were deeply anesthetized with ketamine/xylazine and transcardially perfused with 5–10 mls of 37°C 0.9% sterile saline (pH 7.2) followed by 20–30 mls of 37°C 4% EM-Grade paraformaldehyde (PFA, Electron Microscopy Sciences), 2% EM-Grade glutaraldehyde (GA, Electron Microscopy Sciences), 4 mM calcium chloride ($CaCl_2$, Sigma-Aldrich) in 0.2 M cacodylate buffer (pH 7.4). Brains were postfixed in the same perfusion fixative solution overnight at 4°C. Brains were vibratome sectioned in 0.2 M cacodylate buffer (pH 7.4) at 100 µm. A circular tissue punch (~500 µm diameter) containing the dLGN was microdissected from each section using a blunt-end needle. dLGN sections were washed in 0.2 M cacodylate buffer (pH 7.4) at 4°C on a rotator (4 × 20 min each). Sections were incubated in 20 mM glycine (Sigma-Aldrich) in 0.2 M cacodylate buffer (pH 7.4) at 4°C on a rotator for 30 min, followed by 5 × 20 min washes in 0.2 M cacodylate buffer (pH 7.4) at 4°C on a rotator. dLGN sections were immersed in 0.5 mg/ml diaminobenzidine (DAB) solution, covered with foil, for 30 min at 4°C on a rotator. 10 µl of 0.03% hydrogen peroxide (Sigma-Aldrich) was mixed into the DAB solution (1 ml) and incubated for 10 min at 4°C on a rotator (light protected). The reaction was quenched with 0.2 M cacodylate buffer (pH 7.4) washes (3 × 1 min washes followed by 2 × 20 min washes).

## Tissue preparation for scanning EM

dLGN sections were fixed in 2% osmium tetroxide (Electron Microscopy Sciences) in 0.15 M cacodylate buffer (pH 7.4) for 45 min at room temperature. Sections were reduced with 2.5% potassium ferricyanide (Electron Microscopy Sciences) in 0.15 M cacodylate buffer (pH 7.4) for 45 min at room temperature in the dark, followed by 2 × 10 min washes in double distilled water. Sections were incubated in 1% aqueous thiocarbohydrazide (Electron Microscopy Sciences) for 20 min, followed by 2 × 10 min washes in double distilled water. Samples were fixed in 1% aqueous osmium tetroxide for 45 min at room temperature, followed by 2 × 10 min washes in double distilled water. Sections were postfixed in 1% uranyl acetate (Electron Microscopy Sciences) in 25% ethanol in the dark for 20 min at room temperature and washed and stored in double distilled water overnight. The next day, samples were stained with aqueous lead aspartate at 60°C for 30 min and washed with double-distilled water 2 × 10 min. Tissues were dehydrated in a graded dilution series of 100% ethanol (35%; 50%; 70%; 95%; 100%; 100%; 100% EtOH) for 10 min each at room temperature. Samples were immersed in propylene oxide (Electron Microscopy Sciences) 3 × 10 min at room temperature, followed by a series of epon resin/propylene oxide (812 Epon Resin, Electron Microscopy Sciences) exchanges with increasing resin concentrations (50% resin/50% propylene oxide; 65% resin/35% propylene oxide;

75% resin/25% propylene oxide; 100% resin; 100% resin) for 90 min each. Tissues were transferred to BEEM capsules (Electron Microscopy Sciences) that were filled with 100% resin and polymerized for at least 48 hr at 60°C.

## Transmission EM image acquisition

Plasticized sections were cut at 70 nm with a Histo Jumbo diamond knife (DiATOME) using a Leica UC7 ultramicrotome. Sections were decompressed using chloroform vapor and collected onto 3.05 mm 200 mesh Gilder thin bar hexagonal mesh copper grids (T200H-Cu, Electron Microscopy Sciences). Sections were imaged unstained on a Hitachi HT7700 transmission electron microscope (HT7700, Hitachi High-Tech America, Inc) at 80 kV.

## Eye injections

Intraocular eye injections were performed 1 day before tissue collection. Briefly, mice were anesthetized by inhalant isoflurane, and sterile surgical spring scissors were used to gently part the eyelid to expose the corneoscleral junction. A small hole was made in the eye using a sterile 34-gauge needle and ~0.5 µl of CTB conjugated with Alexa Fluor 488 (CTB-488, Thermo Fisher Scientific, Catalogue Number: C34775) diluted in 0.9% sterile saline was intravitreally pressure-injected into the right eye using a pulled-glass micropipette coupled to a Picospritzer (Parker Hannifin).

## dLGN tissue preparation for STORM imaging

Animals were deeply anesthetized with ketamine/xylazine and transcardially perfused with 5–10 ml of 37°C 0.9% sterile saline followed by 10 mls of room temperature 4% EM-Grade PFA (Electron Microscopy Sciences) in 0.9% saline. Brains were embedded in 2.5% agarose and vibratome sectioned in the coronal plane at 100 µm. From the full anterior–posterior series of dLGN sections (~6–8 sections), we selected the central two sections for staining in all biological replicates. These sections were morphologically consistent with Figures 134–136 (5.07–5.31 mm) of the postnatal day 6 mouse brain from Paxinos's 'Atlas of the developing mouse brain' Academic Press, 2020 (*Paxinos, 2007*). Selected sections were postfixed in 4% PFA for 30 min at room temperature and washed for 30–40 min in 1X PBS. The dLGN was identified by the presence of CTB-488 signals using a fluorescence dissecting microscope. A circular tissue punch (~500 µm diameter) containing the dLGN was microdissected from each section using a blunt-end needle. A small microknife cut was made at the dorsal edge of the dLGN which, together with the CTB-488 signal, enabled us to identify the dLGN orientation during image acquisition.

## Immunohistochemistry

We used a serial-section single-molecule localization imaging approach to prepare samples and collect super-resolution fluorescence imaging volumes as previously described (*Zhang et al., 2023*). dLGN tissue punches were blocked in 10% normal donkey serum (Jackson ImmunoResearch, Catalogue Number: 017-000-121) with 0.3% Triton X-100 (Sigma-Aldrich Inc) and 0.02% sodium azide (Sigma-Aldrich Inc) diluted in 1X PBS for 2–3 hr at room temperature and then incubated in primary antibodies for ~72 hr at 4°C. Primary antibodies used were Rabbit anti-Homer1 (Synaptic Systems, Catalogue Number: 160003, 1:100) to label postsynaptic densities, mouse anti-Bassoon (Abcam, Catalogue Number: AB82958, 1:100) to label presynaptic active zones (AZs), and guinea pig anti-VGluT2 (Millipore, Catalogue Number: AB251-I, 1:100) to label presynaptic vesicles. Following primary antibody incubation, tissues were washed in 1X PBS for 6 × 20 min at room temperature and incubated in secondary antibody solution overnight for ~36 hr at 4°C. The secondary antibodies used were donkey anti-rabbit IgG (Jackson ImmunoResearch, Catalogue Number: 711-005-152, 1:100) conjugated with Dy749P1 (Dyomics, Catalogue Number: 749P1-01) and Alexa Fluor 405 (Thermo Fisher, Catalogue Number: A30000), donkey anti-mouse IgG (Jackson ImmunoResearch, Catalogue Number: 715-005-150, 1:100) conjugated with Alexa Fluor 647 (Thermo Fisher, Catalogue Number: A20006) and Alexa Fluor 405, and donkey anti-guinea pig IgG (Jackson ImmunoResearch, Catalogue Number: 706-005-148, 1:100) conjugated with Cy3b (Cytiva, Catalogue Number: PA63101). Tissues were washed 6 × 20 min in 1X PBS at room temperature after secondary antibody incubation.

## Postfixation, dehydration, and embedding in epoxy resin

Tissue embedding was performed as previously described by *Zhang et al., 2023*. Tissues were postfixed with 3% PFA + 0.1% GA (Electron Microscopy Sciences) in PBS for 2 hr at room temperature and

then washed in 1X PBS for 20 min. To plasticize the tissues for ultrasectioning, the tissues were first dehydrated in a graded dilution series of 100% ethanol (50%/70%/90%/100%/100% EtOH) for 15 min each at room temperature and then immersed in a series of epoxy resin/100% EtOH exchanges (Electron Microscopy Sciences) with increasing resin concentration (25% resin/75% ethanol; 50% resin/50% ethanol; 75% resin/25% ethanol; 100% resin; 100% resin) for 2 hr each. Tissues were transferred to BEEM capsules (Electron Microscopy Sciences) that were filled with 100% resin and polymerized for 16 hr at 70°C.

### Ultrasectioning

Plasticized tissue sections were cut using a Leica UC7 ultramicrotome at 70 nm using a Histo Jumbo diamond knife (DiATOME). Chloroform vapor was used to reduce compression after cutting. For each sample, ~100 sections were collected on a coverslip coated with 0.5% gelatin and 0.05% chromium potassium (Sigma-Aldrich Inc), dried at 60°C for 25 min, and protected from light prior to imaging.

### Imaging chamber preparation

Coverslips were chemically etched in 10% sodium ethoxide for 5 min at room temperature to remove the epoxy resin and expose the dyes to the imaging buffer for optimal photoswitching. Coverslips were then rinsed with ethanol and $dH_2O$. To create fiducial beads for flat-field and chromatic corrections, we mixed 715/755 and 540/560 nm, carboxylate-modified microspheres (Invitrogen, Catalogue Numbers: F8799 and F8809, 1:8 ratio, respectively) to create a high-density fiducial marker and then further diluted the mixture at 1:750 with Dulbecco's PBS to create a low-density bead solution. Both high- and low-density bead solutions were spotted on the coverslip (~0.7 μl each) for flat-field and chromatic aberration correction, respectively. Excess beads were rinsed away with $dH_2O$ for 1–2 min. The coverslip was attached to a glass slide with double-sided tape to form an imaging chamber. The chamber was filled with STORM imaging buffer (10% glucose, 17.5 μM glucose oxidase, 708 nM catalase, 10 mM MEA, 10 mM NaCl, and 200 mM Tris) and sealed with epoxy.

### Imaging setup

Imaging was performed using a custom single-molecule super-resolution imaging system. The microscope contained low (4x/10x air) and high (60x 1.4NA oil immersion) magnitude objectives mounted on a commercial frame (Nikon Ti-U) with back optics arranged for oblique incident angle illumination. We used continuous-wave lasers at 488 nm (Coherent), 561 nm (MPB), 647 nm (MPB), and 750 nm (MPB) to excite Alexa 488, Cy3B, Alexa 647, and Dy749P1 dyes, respectively. A 405-nm cube laser (Coherent) was used to reactivate Dy749P1 and Alexa647 dye photoswitching. The microscope was fitted with a custom pentaband/pentanotch dichroic filter set and a motorized emission filter wheel. The microscope also contained an IR laser-based focus lock system to maintain optimal focus during automatic image acquisition. Images were collected on 640*640-pixel region of an sCMOS camera (ORCA-Flash4.0 V3, Hamamatsu Photonics) with a pixel size of ~155 nm.

### Automated image acquisition

Fiducials and tissue sections on the coverslip were imaged using the low magnification objective (4X) to create a mosaic overview of the specimen. Beads/sections were then imaged at high magnification (60X) to select regions of interest (ROIs) in the Cy3B and Alexa 488 channels. Before final image acquisition, laser intensities and the incident angle were adjusted to optimize photoswitching for STORM imaging and utilize the full dynamic range of the camera for conventional imaging.

Low-density bead images were taken in 16 partially overlapping ROIs. 715/755 nm beads were excited using 750 nm light and images were collected through Dy749P1 and Alexa 647 emission filters. 540/560 nm beads were excited using a 488-nm laser and images were collected through Alexa 647, Cy3B, and Alexa 488 emission filters. These fiducial images were later used to generate a nonlinear warping transform to correct chromatic aberration. Next, ROIs within each tissue section were imaged at conventional (diffraction-limited) resolution in all four-color channels sequentially.

Following conventional image acquisition, a partially overlapping series of nine images was collected in the high-density bead field for all four channels (Dy749P1, Alexa 647, Cy3B, and Alexa 488). These images were later used to perform a flat-field image correction of non-uniform laser illumination across the ROIs. Another round of bead images was taken as described above in a different

ROI of the low-density bead field. These images were later used to confirm the stability of chromatic offsets during imaging. All ROIs within physical sections were then imaged by STORM for Dy749P1 and Alexa 647 channels. Images were acquired using a custom progression of increasing 405 nm laser intensity to control single-molecule switching. 8000 frames of Dy749P1 channel images were collected (60 Hz imaging) followed by 12,000 frames of Alexa 647 channel images (100 Hz). In a second imaging pass, the same ROIs were imaged for Cy3B and Alexa 488 channels, each for 8000 frames (60 Hz).

We imaged the ipsilateral and contralateral ROIs separately in each physical section of the dLGN. For consistency of ROI selection across biological replicates at each age, we identified the dorsal–ventral (DV) axis of the dLGN and selected ROIs within the center (core region) at 2/5 (ipsilateral ROI) and 4/5 (contralateral ROI) of the full DV length.

## Image processing

Single-molecule localization was performed using a previously described DAOSTORM algorithm modified for use with sCMOS cameras (*Babcock et al., 2012*; *Babcock et al., 2019*). Molecule lists were rendered as 8-bit images with 15.5 nm pixel size where each molecule is plotted as an intensity distribution with an area reflecting its localization precision. Low-density fiducial images were used for chromatic aberration correction. We localized 715/755 beads in Dy749P1 and Alexa 647 channels, and 540/560 beads in Alexa 647, Cy3B, and Alexa 488 channels. A third-order polynomial transform map was generated by matching the positions of each bead in all channels to the Alexa 647 channel. The average residual error of bead matching was <15 nm for all channels. The transform maps were applied to both four-color conventional and STORM images. Conventional images were upscaled (by 10X) to match the STORM image size. The method to align serial sections was previously described (*Zhang et al., 2023*). STORM images were first aligned to their corresponding conventional images by image correlation. To generate an aligned 3D image stack from serial sections, we normalized the intensity of all Alexa 488 images and used these normalized images to generate both rigid and elastic transformation matrices for all four-color channels of both STORM and conventional data. The final image stack was then rotated and cropped to exclude incompletely imaged edge areas. To further confirm that the processed region corresponds to the contralateral dLGN, conventional CTB(+) signals of the labeled contralateral projection were thresholded to create a polygonal mask (a convex hull analysis linking the outermost CTB signals in the image volume). The mask was then applied to STORM images to exclude peripheral areas where CTB signals were absent or faint.

## Cell body filter

The aligned STORM images had non-specific labeling of cell bodies in Dy749P1 and Alexa 647 channels corresponding to Homer1 and Bassoon immunolabels. To limit synaptic cluster identification to the neuropil region, we identified cell bodies based on their Dy749P1 signal and excluded these regions from further image processing. STORM images were convolved with a Gaussian function ($\sigma$ = 140 nm) and then binarized using the lower threshold of a two-level Otsu threshold method. We located connected components in the thresholded images and generated a mask based on components larger than e (*Monavarfeshani et al., 2018*) voxels. Because cell body clusters were orders of magnitude larger than synaptic clusters, the cell body filter algorithm was robust to a range of size thresholds. The mask was applied to images of all channels to exclude cell body areas.

## Eye-specific synapse identification and quantification

To correct for minor variance in image intensity across physical sections, we normalized the pixel intensity histogram of each section to the average histogram of all sections. Image histograms were rescaled to make full use of the 8-bit range. Using a two-level Otsu threshold method, the conventional images were thresholded into three classes: a low-intensity background, low-intensity signals above the background representing non-synaptic labeling, and high-intensity signals representing synaptic structures. The conventional images were binarized by the lower two-level Otsu threshold, generating a mask for STORM images to filter out background signals. STORM images were convolved with a Gaussian function ($\sigma$ = 77.5 nm) and thresholded using the higher two-level Otsu threshold. Following thresholding, connected components were identified in three dimensions using MATLAB 'conncomp' function. A watershedding approach was applied to split large clusters that were improperly connected. Clusters were kept for further analysis only if they contained aligned

image information across two or more physical sections. We also removed all edge synapses from our analysis by excluding synapses that did not have blank image data on all adjacent sides.

To distinguish non-specific immunolabeling from true synaptic signals, we quantified two parameters for each cluster: cluster volume and cluster signal density calculated by the ratio of within-cluster pixels with positive signal intensity in the raw STORM images. Two separate populations were identified in 2D histograms plotted from these two parameters. We manually selected the population with higher volumes and signal densities representing synaptic structures. To test the robustness of the manual selection, we performed multiple repeated measurements of the same data and discovered a between-measurement variance of <1% (data not shown).

To identify paired pre- and postsynaptic clusters, we first measured the centroid–centroid distance of each cluster in the Dy749P1 (Homer1) and Alexa 647 (Bassoon) channels to the closest cluster in the other channel. We next quantified the signal intensity of each opposing synaptic channel within a 140-nm shell surrounding each cluster. A 2D histogram was plotted based on the measured centroid–centroid distances and opposing channel signal densities of each cluster. Paired clusters with closely positioned centroids and high intensities of apposed channel signal were identified using the OPTICS algorithm. Retinogeniculate synapses were identified by pairing Bassoon (Alexa 647) clusters with VGluT2 (Cy3B) clusters using the same method as pre/postsynaptic pairing. Synapses from the right eye were identified by pairing VGluT2 clusters with CTB (Alexa 488) clusters. The volume of each cluster reflected the total voxel volume of all connected voxels, and the total signal intensity was a sum of voxel intensity within the volume of the connected voxels.

## Multi-AZ synapse identification and quantification

To determine whether an eye-specific VGluT2 cluster is a mAZ synapse or a sAZ synapse, we measured the number of active zones (defined by individual Bassoon clusters) associated with each VGluT2 cluster in the dataset. A 3D shell was extended 140 nm from the surface voxels of each VGluT2 cluster, and any Bassoon clusters that fell within the shell were considered to be associated with the target VGluT2 cluster. The number of AZs associated with each VGluT2 cluster was then measured. VGluT2 clusters associated with more than one AZ were defined as mAZ synapses, while those associated with only one AZ were defined as sAZ synapses.

Quantification of mAZ and sAZ synapse VGluT2 cluster volume was performed using the 'region-props' function in MATLAB, which provided the voxel size and weighted centroid of each VGluT2 cluster. The search for sAZ synapses adjacent to mAZ synapses (synaptic clustering analysis) was conducted using a similar search approach as for associated Bassoon clusters, with expansion shell sizes ranging from 1 to 4 μm from the surface voxels of each mAZ synapse. The main figures in the study utilized an expansion distance of 1.5 μm. An eye-specific sAZ synapse was considered to be near an mAZ synapse if its weighted centroid fell within the expanded volume.

## Quantification and statistical analysis

Statistical analysis was performed using SPSS. Plots were generated by SPSS or R (ggplot2). Statistical details are presented in the figure legends and . For all measurements in this paper, we analyzed $N = 3$ biological replicates (individual mice) for each genotype (WT and β2KO) at each age (P2, P4, and P8). Cluster densities, synapse AZ number, average VGluT2 cluster volume, and all fractional measurements were presented as mean ± SEM values in paired bar graphs, and statistical analysis was performed using two-tailed paired *T*-tests. Nonparametric Kolmogorov–Smirnov tests were used in all cumulative histogram comparisons. We used a linear mixed model to compare VGluT2 cluster volumes (*Figure 3*; *Figure 3—figure supplement 1*) and distance measurements (*Figure 5*; *Figure 5—figure supplement 1*). For VGluT2 cluster volume comparisons, the age or eye-of-origin was the fixed main factor and biological replicate IDs were nested random factors. In distance measurement comparisons, the mAZ synapse AZ number was the fixed main factor and biological replicate IDs were nested random factors. Effect sizes were calculated using Cohen's d for parametric tests and epsilon squared for nonparametric tests. To address multiple testing, we applied FDR correction using the Benjamini–Hochberg method with $\alpha = 0.05$ separately within each experimental condition (P2-WT, P2-β2KO, P4-WT, P4-β2KO, P8-WT, and P8-β2KO). 20–34 measurements (varying by age and genotype) were corrected within each of the six experimental conditions, resulting in condition-specific FDR-adjusted p-values presented in *Supplementary file 2*. In violin plots, each violin showed the distribution of

grouped data from all biological replicates from the same condition. Each black dot represents the median value of each biological replicate, and the horizontal black line represents the group median. Black lines connect measurements of CTB(+) and CTB(−) populations from the same biological replicate. Asterisks in all figures indicate statistical significance after FDR correction: *p(adj) < 0.05. For VGluT2 volume comparisons, we calculated 5/95% confidence intervals based on the linear mixed model. 5/95% confidence intervals in synapse densities, AZ numbers, and distances were calculated for paired or unpaired data comparisons using SPSS. All statistical results are listed in *Supplementary file 2*. All conclusions we draw in the main text from the data have corresponding confidence intervals listed in both the figure legends and *Supplementary file 3*.

## Acknowledgements

We thank Drs. Michael C Crair (Yale University), Eric M Ullian (University of California, San Francisco), and Joshua H Singer (University of Maryland) for generously sharing the β2KO, *SLC6A4*$^{Cre}$, and *ROSA26*$^{LSL-Matrix-dAPEX2}$ mouse lines used in this work. We are grateful to Tim Maugel in the Laboratory for Biological Ultrastructure (UMD) for his assistance with transmission electron microscopy experiments. The research was funded by National Institutes of Health grant DP2MH125812 (CMS).

## Additional information

### Funding

| Funder | Grant reference number | Author |
| --- | --- | --- |
| National Institutes of Health | DP2MH125812 | Colenso M Speer |

The funders had no role in study design, data collection, and interpretation, or the decision to submit the work for publication.

### Author contributions

Chenghang Zhang, Conceptualization, Resources, Data curation, Software, Formal analysis, Validation, Investigation, Visualization, Methodology, Writing – original draft, Project administration, Writing – review and editing; Tarlan Vatan, Resources, Data curation, Validation, Investigation, Visualization, Methodology, Writing – review and editing; Colenso M Speer, Conceptualization, Resources, Data curation, Software, Formal analysis, Supervision, Funding acquisition, Validation, Investigation, Visualization, Methodology, Writing – original draft, Project administration, Writing – review and editing

### Author ORCIDs

Chenghang Zhang (iD) https://orcid.org/0000-0003-1346-2186
Colenso M Speer (iD) https://orcid.org/0000-0002-3076-7072

### Ethics

All experimental procedures were performed in accordance with an animal study protocol approved by the Institutional Animal Care and Use Committee (IACUC) at the University of Maryland (Protocol # R-JUL-23-24).

Joint Public Review: https://doi.org/10.7554/eLife.91431.5.sa1
Author response https://doi.org/10.7554/eLife.91431.5.sa2

## Additional files

### Supplementary files

MDAR checklist

Supplementary file 1. mAZ and sAZ synapses numbers for all biological replicates.

Supplementary file 2. Statistical analyses for all figures.

Supplementary file 3. Confidence interval analyses.

### Data availability

All Matlab and Python code used in the work is available on GitHub (https://github.com/SpeerLab/Aligned_data_analysis_SynapseClustering; copy archived at *Zhang and Speer, 2025*). Raw STORM images of the full data are available on the open-access Brain Imaging Library https://doi.org/10.35077/g.1164.

The following dataset was generated:

| Author(s) | Year | Dataset title | Dataset URL | Database and Identifier |
|-----------|------|---------------|-------------|-------------------------|
| Zhang C, Yadav S, Speer CM | 2024 | The synaptic basis of activity-dependent eye-specific competition | https://doi.org/10.35077/g.1164 | Brain Image Library, 10.35077/g.1164 |

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
