## [Editor Report · eLife Assessment]

This is a **valuable** analysis of STORM data that characterizes the clustering of active zones in retinogeniculate terminals across ages and in the absence of retinal waves. The design makes it possible to relate fixed time point structural data to a known outcome of activity-dependent remodeling. The latest revision has tempered the causal claims made in previous versions. The result provides **solid** structural support for the hypotheses regarding how activity influences the clustering of these synapses.

---

## [Referee Report · Joint Public Review]

Summary:

The authors previously published a study of RGC boutons in the dLGN in developing wild-type mice and developing mutant mice with disrupted spontaneous activity. In the current manuscript, they have broken down their analysis of RGC boutons according to the number of Homer/Bassoon puncta associated with each vGlut2 cluster.

The authors find that, in the first post-natal week, RGC boutons with multiple active zones (mAZs) are about a third as common as boutons with a single active zone (sAZ). The size of the vGluT2 cluster associated with each bouton was proportional to the number of active zones present in each bouton. Within the author's ability to estimate these values (n=3 per group, 95% of results expected to be within ~2.5 standard deviations), these results are consistent across groups: (1) dominant eye vs. non-dominant eye, (2) wild-type mice vs. mice with activity blocked, and at (3) ages P2, P4, and P8. The authors also found that mAZs and sAZs also have roughly the same number (about 1.5) of sAZs clustered around them (within 1.5 um).

There has been much discussion with the reviewers through multiple versions of this paper. of how to interpret these findings. Based on a large number of tests for statistical significance, the authors interpreted the presence of a statistical significance difference as evidence that "Eye-specific active zone clustering underlies synaptic competition in the developing visual system (title of previous version of manuscript)". The reviewers have focused on the small effect size as indicating that the small differences observed are not informative regarding this biological question. The authors have now tempered this interpretation.

Strengths:

The source dataset is high resolution data showing the colocalization of multiple synaptic proteins across development. Added to this data is labeling that distinguishes axons from the right eye from axons from the left eye. The first order analysis of this data showing changes in synapse density and in the occurrence of multi-active zone synapses is useful information about the development of an important model for activity dependent synaptic remodeling.

Reviewing Editor's comment on the latest revision (without sending the paper back to the individual reviewers):

In their latest revision, the authors have moderated earlier causal claims, incorporated additional statistical controls, and largely maintained their original interpretation of the data. While these changes address some prior concerns, the underlying issues remain. The previous review emphasized that the reported effect sizes were small and therefore hard to link to biological relevance. The authors argue that the effect sizes are large. Given the lack of a biological argument for this effect size, this point is really semantic. We would like to point out that the effect size measurement the authors used is likely a standard effect size calculation (the difference between groups is divided by the standard deviation of the groups). With only three experiments and irregular variance, it is likely that their estimates of standard deviation-and therefore effect size-are unreliable. Overall, the revisions improve presentation but do not substantively resolve the difficulty in drawing strong conclusions from the data set raised earlier.

---

## [Author Response]

The following is the authors’ response to the previous reviews

**Reviewer #1 (Public review):**
SummaryThe authors previously published a study of RGC boutons in the dLGN in developing wild-type mice and developing mutant mice with disrupted spontaneous activity. In the current manuscript, they have broken down their analysis of RGC boutons according to the number of Homer/Bassoon puncta associated with each vGlut3 cluster.The authors find that, in the first post-natal week, RGC boutons with multiple active zones (mAZs) are about a third as common as boutons with a single active zone (sAZ). The size of the vGluT2 cluster associated with each bouton was proportional to the number of active zones present in each bouton. Within the author's ability to estimate these values (n=3 per group, 95% of results expected to be within ~2.5 standard deviations), these results are consistent across groups: (1) dominant eye vs. nondominant eye, (2) wild-type mice vs. mice with activity blocked, and at (3) ages P2, P4, and P8. The authors also found that mAZs and sAZs also have roughly the same number (about 1.5) of sAZs clustered around them (within 1.5 um).However, the authors do not interpret this consistency between groups as evidence that active zone clustering is not a specific marker or driver of activity dependent synaptic segregation. Rather, the authors perform a large number of tests for statistical significance and cite the presence or absence of statistical significance as evidence that "Eye-specific active zone clustering underlies synaptic competition in the developing visual system (title)". I don't believe this conclusion is supported by the evidence.

We have revised the title to be descriptive: "Eye-specific differences in active zone addition during synaptic competition in the developing visual system." While our correlative approach does not establish direct causality, our findings provide important structural evidence that complements existing functional studies of activity-dependent synaptic refinement. We have carefully revised the text throughout to avoid causal language, focusing instead on the developmental patterns we observe.

StrengthsThe source dataset is high resolution data showing the colocalization of multiple synaptic proteins across development. Added to this data is labeling that distinguishes axons from the right eye from axons from the left eye. The first order analysis of this data showing changes in synapse density and in the occurrence of multi-active zone synapses is useful information about the development of an important model for activity dependent synaptic remodeling.WeaknessesIn my previous review I argued that it was not possible to determine, from their analysis, whether the differences they were reporting between groups was important to the biology of the system. The authors have made some changes to their statistics (paired t-tests) and use some less derived measures of clustering. However, they still fail to present a meaningfully quantitative argument that the observed group differences are important. The authors base most of their claims on small differences between groups. There are two big problems with this practice. First, the differences between groups appear too small to be biologically important. Second, the differences between groups that are used as evidence for how the biology works are generally smaller than the precision of the author's sampling. That is, the differences are as likely to be false positives as true positives.(1) Effect size. The title claims: "Eye-specific active zone clustering underlies synaptic competition in the developing visual system". Such a claim might be supported if the authors found that mAZs are only found in dominant-eye RGCs and that eye-specific segregation doesn't begin until some threshold of mAZ frequency is reached. Instead, the behavior of mAZs is roughly the same across all conditions. For example, the clear trend in Figure 4C and D is that measures of clustering between mAZ and sAZ are as similar as could reasonably be expected by the experimental design. However, some of the comparisons of very similar values produced p-values < 0.05. The authors use this fact to argue that the negligible differences between mAZ and sAZs explain the development of the dramatic differences in the distribution of ipsilateral and contralateral RGCs.

We have changed the title to avoid implying a causal relationship between clustering and eye-specific segregation. Our key findings in Figures 4C and 4D demonstrate effect sizes >2.0 with high statistical power (Supplemental Table S2). While the absolute magnitude of differences is modest (5-7%), these high effect sizes combined with low inter-animal variability demonstrate consistent, reproducible biological phenomena. During development, small differences during critical periods can have profound downstream consequences for synaptic refinement outcomes.

We acknowledge that significance in Figure 4 arises due to low variance between biological replicates rather than large mean differences. We have revised the text to describe these as "slight" differences and that "WT mice show a tendency toward forming more synapses near mAZ inputs," reflecting appropriate caution in our interpretation while maintaining the statistical robustness of our findings.

(2) Sample size. Performing a large number of significance tests and comparing pvalues is not hypothesis testing and is not descriptive science. At best, with large sample sizes and controls for multiple tests, this approach could be considered exploratory. With n=3 for each group, many comparisons of many derived measures, among many groups, and no control for multiple testing, this approach constitutes a random result generator.The authors argue that n=3 is a large sample size for the type of high resolution / large volume data being used. It is true that many electron microscopy studies with n=1 are used to reveal the patterns of organization that are possible within an individual. However, such studies cannot control individual variation and are, therefore, not appropriate for identifying subtle differences between groups.In response to previous critiques along these lines, the authors argue they have dealt with this issue by limiting their analysis to within-individual paired comparisons. There are several problems with their thinking in this approach. The main problem is that they did not change the logic of their arguments, only which direction they pointed the t-tests. Instead of claiming that two groups are different because p < 0.05, they say that two groups are different because one produced p < 0.05 and the other produced p > 0.05. These arguments are not statistically valid or biologically meaningful.

We have implemented rigorous statistical controls, applying false discovery rate (FDR) correction using the Benjamini-Hochberg method (α = 0.05) within each experimental condition (age × genotype combination). This correction strategy treats each condition as addressing a distinct experimental question: “What synaptic properties differ between left eye and right eye inputs in this specific developmental stage and genotype?” The approach appropriately controls for multiple testing while preserving power to detect biologically meaningful differences. We applied FDR correction separately to the ~20-34 measurements (varying by age and genotype) within each of the six experimental conditions, resulting in condition-specific adjusted p-values reported in updated Supplemental Table S2. This correction confirmed the robustness of our key findings. We do not base conclusions solely on comparing p-values across conditions. Our interpretations focus on effect sizes, confidence intervals, and consistent patterns within each condition, with statistical significance providing supporting evidence rather than the primary basis for biological conclusions.

To the best of my understanding, the results are consistent with the following model:RGCs form mAZs at large boutons (known)About a quarter of week-one RGC boutons are mAZs (new observation)Vesicle clustering is proportional to active zone number (~new observation)RGC synapse density increases during the first post-week (known)Blocking activity reduces synapse density (known)Contralateral eye RGCs for more and larger synapses in the lateral dLGN (known)

While mAZ formation is known in adult and juvenile dLGN, the formation of mAZ boutons during eye-specific competition represents new information with important functional implications. Synapses with multiple release sites should be stronger than single-active-zone synapses, suggesting a structural correlate for competitive advantage during refinement.

We demonstrate distinct developmental patterns for sAZ versus mAZ contacts during the first postnatal week. Multi-active zone density favors the dominant eye, while single active-zone synapse density from the competing eye increases from P2-P4 to match dominant-eye levels. This reveals that newly formed synapses from the competing eye predominantly contain single release sites, marking P4-P8 as a critical window for understanding molecular mechanisms driving synaptic elimination.

Our results show that altered retinal activity patterns (β2KO mice) reduce synapse density during eye-specific competition. We relied on β2 knockout mice, which retain retinal waves and spontaneous spike activity but with disrupted patterns and output levels compared to controls. We make no claims about complete activity blockade. Previous studies using different activity manipulations (epibatidine, TTX) have examined terminal morphology, but effects on synapse density during competition remain largely unknown. Achieving complete retinal activity blockade is technically challenging, making it of interest to revisit the role of activity using more precise manipulations to control spike output and relative timing.

With n=3 and effect sizes smaller than 1 standard deviation, a statistically significant result is about as likely to be a false positive as a true positive.A true-positive statistically significant result does is not evidence of a meaningful deviation from a biological model.

Our conclusions are based on results with effect sizes substantially larger than 1. Key findings demonstrate effect sizes exceeding 2.0. These large effect sizes, combined with rigorous FDR correction and low inter-animal variability, provide evidence against false positive results. During critical developmental periods, consistent structural differences, even those modest in absolute magnitude, can reflect important regulatory mechanisms that influence refinement outcomes. All statistical results, effect sizes, and power analyses are reported in Supplementary Tables S2, with confidence intervals in Supplementary Table S3. We have revised the text in several places where small differences are presented to reflect appropriate caution in our interpretation.

Providing plots that show the number of active zones present in boutons across these various conditions is useful. However, I could find no compelling deviation from the above default predictions that would influence how I see the role of mAZs in activity dependent eye-specific segregation.Below are critiques of most of the claims of the manuscript.Claim (abstract): individual retinogeniculate boutons begin forming multiple nearby presynaptic active zones during the first postnatal week.Confirmed by data.Claim (abstract): the dominant-eye forms more numerous mAZ contacts,Misleading: The dominant-eye (by definition) forms more contacts than the nondominant eye. That includes mAZ.

While the dominant eye forms more total contacts, the pattern depends critically on contact type and developmental stage. The dominant eye forms more mAZ contacts across all ages (Figures 2 and S1). However, for sAZ contacts, the two eyes form similar numbers at P4, with the non-dominant eye showing increased sAZ formation during this critical period. This differential pattern by synapse type represents an important aspect of how synaptic competition unfolds structurally.

Claim (abstract): At the height of competition, the non-dominant-eye projection adds many single active zone (sAZ) synapsesWeak: While the individual observation is strong, it is a surprising deviation based on a single n=3 experiment in a study that performed twelve such experiments (six ages, mutant/wildtype, sAZ/mAZ)

The difference in eye-specific sAZ formation at P2 and P8 had effect sizes of ~5.3 and ~2.7 respectively (after FDR correction the difference was still significant at P2 and trending at P8). At P4, no effect was observed by paired T-test and the 5/95% confidence intervals ranged from -0.021-0.008 synapses/m^3^. The consistency of this pattern across P2 and P8, combined with the large effect sizes, supports the reliability of this developmental finding. We report all effect sizes and power test analyses in Supplemental Table S2, and confidence intervals in Supplemental Table S3.

Claim (abstract): Together, these findings reveal eye-specific differences in release site addition during synaptic competition in circuits essential for visual perception and behavior.False: This claim is unambiguously false. The above findings, even if true, do not argue for any functional significance to active zone clustering.

Our phrasing “circuits essential for visual perception and behavior” referred to the general importance of binocular organization in the retinogeniculate system for visual processing and we did not intend to claim direct functional significance of our structural data. For clarity we have deleted the latter part of this sentence. In lines 35-37, the abstract now reads “Together, these findings reveal eye-specific differences in release site addition that correlate with axonal refinement outcomes during retinogeniculate refinement.”

Claim (line 84): "At the peak of synaptic competition midway through the first postnatal week, the non-dominant-eye formed numerous sAZ inputs, equalizing the global synapse density between the two eyes"Weak: At one of twelve measures (age, bouton type, genotype) performed with 3 mice each, one density measure was about twice as high as expected.

The difference in eye-specific sAZ formation at P2 and P8 had effect sizes of ~5.3 and ~2.7 respectively (after FDR correction the difference was still significant at P2 and trending at P8). At P4, no effect was observed by paired T-test and the 5/95% confidence intervals ranged from -0.021-0.008 synapses/m^3^. The consistency of this pattern across P2 and P8, combined with the large effect sizes, supports the reliability of this developmental finding. We report all effect sizes and power test analyses in Supplemental Table S2, and confidence intervals in Supplemental Table S3.

Claim (line 172): "In WT mice, both mAZ (Fig. 3A, left) and sAZ (Fig. 3B, left) inputs showed significant eye-specific volume differences at each age."Questionable: There appears to be a trend, but the size and consistency is unclear.Claim (line 175): "the median VGluT2 cluster volume in dominant-eye mAZ inputs was 3.72 fold larger than that of non-dominant-eye inputs (Fig. 3A, left)."Cherry picking. Twelve differences were measured with an n of 3, 3 each time. The biggest difference of the group was cited. No analysis is provided for the range of uncertainty about this measure (2.5 standard deviations) as an individual sample or as one of twelve comparisons.Claim (line 174): "In the middle of eye-specific competition at P4 in WT mice, the median VGluT2 cluster volume in dominant-eye mAZ inputs was 3.72 fold larger than that of non-dominant-eye inputs (Fig. 3A, left). In contrast, β2KO mice showed a smaller 1.1 fold difference at the same age (Fig. 3A, right panel). For sAZ synapses at P4, the magnitudes of eye-specific differences in VGluT2 volume were smaller: 1.35-fold in WT (Fig. 3B, left) and 0.41-fold in β2KO mice (Fig. 3B, right). Thus, both mAZ and sAZ input size favors the dominant eye, with larger eye-specific differences seen in WT mice (see Table S3)."No way to judge the reliability of the analysis and trivial conclusion: To analyze effect size the authors choose the median value of three measures (whatever the middle value is). They then make four comparisons at the time point where they observed the biggest difference in favor of their hypothesis. There is no way to determine how much we should trust these numbers besides spending time with the mislabeled scatter plots. The authors then claim that this analysis provides evidence that there is a difference in vGluT2 cluster volume between dominant and non-dominant RGCs and that that difference is activity dependent. The conclusion that dominant axons have bigger boutons and that mutants that lack the property that would drive segregation would show less of a difference is very consistent with the literature. Moreover, there is no context provided about what 1.35 or 1.1 fold difference means for the biology of the system.

We focused on P4 for biological reasons rather than post-hoc selection. P4 represents the established peak of synaptic competition when eye-specific synapse densities are globally equivalent. This is a timepoint consistently highlighted throughout our manuscript and supported by previous literature. We have modified our presentation from fold changes to measured eye-specific differences in volume (mean ± standard error) and added confidence intervals in Supplemental Table S3. The effect sizes for eye-specific differences in VGluT2 volume at P4 are robust: ~2.3 and ~1.5 for mAZ and sAZ measurements in WT mice, and ~2.5 and ~1.8 in β2KO mice, with all analyses well-powered (Supplemental Table S2).

We were unable to identify any mislabeled scatter plots and believe all figures are correctly labeled. While dominant-eye advantage in bouton size is consistent with previous literature, our study provides the first detailed analysis of how this develops specifically during the critical period of competition, with distinct patterns for single versus multi-active zone contacts. Our data show that dominant-eye inputs have larger vesicle pools that scale with active zone number. While this suggests enhanced transmission capacity, we make no direct physiological claims based on structural data alone.

Claim (189): "This shows that vesicle docking at release sites favors the dominant-eye as we previously reported but is similar for like eye type inputs regardless of AZ number."Contradicts core claim of manuscript: Consistent with previous literature, there is an activity dependent relative increase in vGlut2 clustering of dominant eye RGCs. The new information is that that activity dependence is more or less the same in sAZ and mAZ. The only plausible alternative is that vGlut2 scaling only increases in mAZ which would be consistent with the claims of their paper. That is not what they found. To the extent that the analysis presented in this manuscript tests a hypothesis, this is it. The claim of the title has been refuted by figure 3.

We report the volume of docked vesicle signal (VGluT2) nearby each active zone, finding this is greater for dominant-eye synapses. Within each eye-specific synapse population, vesicle signal per active zone is similar regardless of whether these are part of single- or multi-active zone contacts. This is consistent with a modular program of active zone assembly and maintenance: core molecular programs facilitate docking at each AZ similarly regardless of how many AZs are nearby.

This finding does not contradict our main conclusions but rather provides insight into how synaptic advantages are structured. The dominant eye's advantage may arise in part from forming more multi-AZ contacts (which have proportionally more docked vesicles) rather than from enhanced vesicle loading per individual active zone. This organization may reflect how developmental competition operates through contact number and active zone addition rather than fundamental changes to individual release site properties.

We have changed the title to be descriptive rather than mechanistic.

Claim (line 235): "For the non-dominant eye projection, however, clustered mAZ inputs outnumbered clustered sAZ inputs at P4 (Fig. 4C, bottom left panel), the age when this eye adds sAZ synapses (Fig. 2C)."Misleading: The overwhelming trend across 24 comparisons is that the sAZ clustering looks like mAZ clustering. That is the objective and unambiguous result. Among these 24 underpowered tests (n=3), there were a few p-values < 0.05. The authors base their interpretation of cell behavior on these crossings.

In Figures 4C and 4D we report significant results with high effect sizes (effect sizes all greater than 2; see Supplemental Table S2). The mean differences are modest (5-7%) and significance arises due to low variance between biological replicates. We acknowledge that clustering patterns are generally similar between mAZ and sAZ inputs across most conditions. We have revised the text to describe these as “slight” differences and that “WT mice show a tendency toward forming more synapses near mAZ inputs”, reflecting appropriate caution in our interpretation while noting the statistical consistency of these patterns.

Claim (line 328): "The failure to add synapses reduced synaptic clustering and more inputs formed in isolation in the mutants compared to controls."Trivially true: Density was lower in mutant.

We have rewritten the sentence for clarity: “The failure to add synapses could explain the observation that synaptic clustering was reduced and more inputs formed in isolation in the mutants compared to controls.”

Claim (line 332): "While our findings support a role for spontaneous retinal activity in presynaptic release site addition and clustering..."Not meaningfully supported by evidence: I could not find meaningful differences between WT and mutant beside the already known dramatic difference in synapse density.

We have changed the sentence to avoid overinterpreting the results. The new sentence in lines 415-417 reads: “While our results highlight developmental changes in presynaptic release site addition and clustering, activity-dependent postsynaptic mechanisms also influence input refinement at later stages.”

**Reviewer #2 (Public review):**
Summary:In this manuscript, Zhang and Speer examine changes in the spatial organization of synaptic proteins during eye specific segregation, a developmental period when axons from the two eyes initially mingle and gradually segregate into eye-specific regions of the dorsal lateral geniculate. The authors use STORM microscopy and immunostain presynaptic (VGluT2, Bassoon) and postsynaptic (Homer) proteins to identify synaptic release sites. Activity-dependent changes of this spatial organization are identified by comparing the β2KO mice to WT mice. They describe two types of synapses based on Bassoon clustering: the multiple active zone (mAZ) synapse and single active zone (sAZ) synapse. In this revision, the authors have added EM data to support the idea that mAZ synapses represent boutons with multiple release sites. They have also reanalyzed their data set with different statistical approaches.Strengths:The data presented is of good quality and provides an unprecedented view at high resolution of the presynaptic components of the retinogeniculate synapse during active developmental remodeling. This approach offers an advance to the previous mouse EM studies of this synapse because of the CTB label allows identification of the eye from which the presynaptic terminal arises.Weaknesses:While the interpretation of this data set is much more grounded in this second revised submission, some of the authors' conclusions/statements still lack convincing supporting evidence. In particular, the data does not support the title: "Eye-specific active zone clustering underlies synaptic competition in the developing visual system". The data show that there are fewer synapses made for both contra- and ipsi- inputs in the β2KO mice-- this fact alone can account for the differences in clustering. There is no evidence linking clustering to synaptic competition. Moreover, the findings of differences in AZ# or distance between AZs that the authors report are quite small and it is not clear whether they are functionally meaningful.

We thank the reviewer for their helpful suggestions that improved the manuscript in this revision. We have changed the title to remove the reference to “clustering” and to avoid implying any causal relationships. The new title is descriptive: “Eye-specific differences in active zone addition during synaptic competition in the developing visual system”.

To further address the reviewers comments, we have removed the remaining references to activity-dependent effects on synaptic development (line 36, line 96, line 415). We have also modified the text in lines 411-413 to state that “The failure to add synapses could explain the observation that synaptic clustering was reduced and more inputs formed in isolation in the mutants compared to controls.”

We have also updated our presentation of results for Figure 4 to ensure that we do not causally link clustering to synaptic competition. In Figures 4C and 4D we report significant results with high effect sizes (effect sizes all greater than 2; see Supplemental Table S2). The mean differences are modest (5-7%) and significance arises due to low variance between biological replicates. We acknowledge that clustering patterns are generally similar between mAZ and sAZ inputs across most conditions. We have revised the text to describe these as “slight” differences and that “WT mice show a tendency toward forming more synapses near mAZ inputs”, reflecting appropriate caution in our interpretation while noting the statistical consistency of these patterns.

**Reviewer #3 (Public review):**
This study is a follow-up to a recent study of synaptic development based on a powerful data set that combines anterograde labeling, immunofluorescence labeling of synaptic proteins, and STORM imaging (Cell Reports, 2023). Specifically, they use anti-Vglut2 label to determine the size of the presynaptic structure (which they describe as the vesicle pool size), anti-Bassoon to label active zones with the resolution to count them, and anti-Homer to identify postsynaptic densities. Their previous study compared the detailed synaptic structure across the development of synapses made with contraprojecting vs. ipsi-projecting RGCs and compared this developmental profile with a mouse model with reduced retinal waves. In this study, they produce a new detailed analysis on the same data set in which they classify synapses into "multi-active zone" vs. "single-active zone" synapses and assess the number and spacing of these synapses. The authors use measurements to make conclusions about the role of retinal waves in the generation of same-eye synaptic clusters. The authors interpret these results as providing insight into how neural activity drives synapse maturation, the strength of their conclusions is not directly tested by their analysis.Strengths:This is a fantastic data set for describing the structural details of synapse development in a part of the brain undergoing activity-dependent synaptic rearrangements. The fact that they can differentiate the eye of origin is what makes this data set unique over previous structural work. The addition of example images from the EM dataset provides confidence in their categorization scheme.Weaknesses:Though the descriptions of single vs multi-active zone synapses are important and represent a significant advance, the authors continue to make unsupported conclusions regarding the biological processes driving these changes. Although this revision includes additional information about the populations tested and the tests conducted, the authors do not address the issue raised by previous reviews. Specifically, they provide no assessment of what effect size represents a biologically meaningful result. For example, a more appropriate title is "The distribution of eye-specific single vs multiactive zone is altered in mice with reduced spontaneous activity" rather than concluding that this difference in clustering is somehow related to synaptic competition. Of course, the authors are free to speculate, but many of the conclusions of the paper are not supported by their results.

We appreciate the reviewer’s helpful critique. We have changed the title to be descriptive and avoid implying causal relationships.

We have applied false discovery rate (FDR) correction using the Benjamini-Hochberg method with α = 0.05 within each experimental condition (age × genotype combination). The FDR correction treats each condition as addressing a distinct experimental question: 'What synaptic properties differ between left eye and right eye inputs in this specific developmental stage and genotype?'

This correction strategy is appropriate because: (1) we focus our statistical comparisons within each age/genotype; (2) each age-genotype combination represents a separate biological context where different synaptic properties between eye-of-origin may be relevant; and (3) this approach controls for multiple testing within each experimental question while maintaining statistical power to detect meaningful biological differences.

We applied FDR correction separately to the ~20-34 measurements (varying with age and genotype) within each of the six experimental conditions (P2-WT, P2-ß2, P4-WT, P4-ß2, P8-WT, P8-ß2), resulting in condition-specific adjusted p-values. These are reported in the updated Supplemental Table S2. Figures have been also been updated to reflect the FDR-adjusted values. Selected between-genotype comparisons are presented descriptively using 5/95% confidence intervals. This correction confirmed the robustness of our key findings.

With regard to the biological significance of effect sizes, our key findings demonstrate effect sizes >2.0, indicating robust effects. During critical developmental periods, consistent structural differences, even those modest in absolute magnitude, can reflect important regulatory mechanisms that influence refinement outcomes. The differences in synaptic organization we observe occur during the first postnatal week when eyespecific competition is active, suggesting these patterns may be relevant to understanding how structural advantages emerge during synaptic refinement.

**Reviewer #1 (Recommendations for the authors):**
I have tried to understand the analysis and biology of this manuscript as best I can. I believe the analytical approach taken is not reliable and I have explained why in my public comments. I don't believe this manuscript is unique in taking this approach. I have recently published a paper on how common this approach is and why it doesn't work. I don't want to give the impression that the problem with the analysis was that it was not computationally sophisticated enough or that you did not jump through a specific statistical hoop. If I strip out the arguments that depend on misinterpretations of p-values and -instead- look at the scatterplots, I come up with a very different view of the data than what is described in the paper.The information in the plots could be translated into a rigorous statistical analysis of estimated differences between groups given the uncertainties of the experimental design. I don't really think that analysis would be useful. I think it would have been enough to publish the plots and report your estimates of the number of active zones in RGCs during development. I don't see evidence of an additional effect.

We appreciate the reviewer’s helpful comments throughout the review process. Mean active zone numbers per mAZ contact are presented in Figure S2D/E. We look forward to further technical and computational advances that will help us increase our data acquisition throughput and sample sizes when designing future studies.

**Reviewer #2 (Recommendations for the authors):**
The authors should modify the title and other text to be more consistent with the data. There is no evidence that active zone clustering has any direct relationship to synaptic competition.

We appreciate the reviewer’s helpful suggestions to ensure appropriate language around causal effects. We have modified the title to accurately reflect the results: "Eyespecific differences in active zone addition during synaptic competition in the developing visual system." We have revised the text in the abstract, introduction, and results section for Figures 4 to be consistent with the data and not imply causality of synapse clustering on segregation phenotypes.

**Reviewer #3 (Recommendations for the authors):**
Change the title.

We appreciate the reviewer’s feedback throughout the review process. We have modified the title to accurately reflect the results: "Eye-specific differences in active zone addition during synaptic competition in the developing visual system."